# The highly rugged yet navigable regulatory landscape of the bacterial transcription factor TetR

Cauã Antunes Westmann [1,2], Leander Goldbach[1,2] & Andreas Wagner [1,2,3] ✉

Transcription factor binding sites (TFBSs) are important sources of evolutionary innovations. Understanding how evolution navigates the sequence space of such sites can be achieved by mapping TFBS adaptive landscapes. In such a landscape, an individual location corresponds to a TFBS bound by a transcription factor. The elevation at that location corresponds to the strength of transcriptional regulation conveyed by the sequence. Here, we develop an in vivo massively parallel reporter assay to map the landscape of bacterial TFBSs. We apply this assay to the TetR repressor, for which few TFBSs are known. We quantify the strength of transcriptional repression for 17,765 TFBSs and show that the resulting landscape is highly rugged, with 2092 peaks. Only a few peaks convey stronger repression than the wild type. Non-additive (epistatic) interactions between mutations are frequent. Despite these hallmarks of ruggedness, most high peaks are evolutionarily accessible. They have large basins of attraction and are reached by around 20% of populations evolving on the landscape. Which high peak is reached during evolution is unpredictable and contingent on the mutational path taken. This in-depth analysis of a prokaryotic gene regulator reveals a landscape that is navigable but much more rugged than the landscapes of eukaryotic regulators.

Mapping the relationship between genetic and phenotypic variation is crucial to understand the evolutionary process. Genotype-phenotype maps[1,2] are widely used to study this relationship by connecting a potentially large genotype space to phenotypic traits, such as gene expression levels, RNA secondary structures, or enzymatic activity[3–7]. If a trait is a scalar quantity, it can define an elevation at each coordinate (genotype). A special case of the resulting landscape is a fitness or adaptive landscape, in which the trait is an organism's fitness[1,8–10]. Evolution on such a landscape can be conceptualized as a hill-climbing process, in which populations are driven towards high-fitness genotypes by natural selection[1,8,9]. A smooth and single-peaked adaptive landscape facilitates the discovery of high-fitness genotypes. Such a landscape is highly *navigable*[4], because selection can lead a population to the global peak through multiple small "uphill" mutations. In contrast, a rugged landscape[11] can hinder progress towards the highest peak, trapping populations on local suboptimal peaks with low fitness[10–12].

New patterns of gene regulation are important sources of biological innovations[4,13–15]. Transcriptional regulation is mainly controlled by transcription factors (TFs) that bind short DNA sequences known as transcription factor-binding sites (TFBSs). Such binding helps to regulate—activate or repress—gene expression[16,17]. Therefore, mutational changes in TFBSs can play important roles in development, disease, and evolution of novel traits[4,13–15]. Despite numerous theoretical studies exploring the molecular origins and evolution of these elements[18–27], comprehensive and high-throughput empirical research addressing these questions only began about a decade ago[28–40]. Additionally, while extensive in vitro data on the binding of eukaryotic

[1]Department of Evolutionary Biology and Environmental Studies, University of Zurich, Winterthurerstrasse 190, Zurich CH-8057, Switzerland. [2]Swiss Institute of Bioinformatics, Quartier Sorge-Batiment Genopode, 1015 Lausanne, Switzerland. [3]The Santa Fe Institute, Santa Fe, NM 87501, USA. ✉e-mail: andreas.wagner@ieu.uzh.ch

TFs to their binding sites have advanced our understanding of eukaryotic genotype-phenotype maps of gene regulation[4,30,41–45], the topography of these landscapes remains largely unexplored in prokaryotes. The few studies that do examine the relationships between TFBS sequences and gene regulation in bacteria primarily focus on the mechanisms of gene regulation rather than the evolution of gene regulation[28,31,33,46–49].

To bridge this knowledge gap, we map a comprehensive adaptive landscape of TFBS variants for bacteria. We adopt the tetracycline repressor (TetR[50–52]) from the *tet* operon encoded in the Tn10 transposon[51,53] as our model system. TetR is a well-studied TF with only two known cognate binding sites, *tetO1* and *tetO2*[50,52,54]. In the absence of the antibiotic tetracycline, TetR binds independently to *tetO1* and *tetO2*[55–60], repressing its own transcription and the expression of the *tetA* gene, which encodes an antibiotic efflux pump[61]. We focus here on *tetO2*, because it affects the regulation of the *tet* genes more strongly[55–57,62,63]. We refer to it as our wild-type binding site.

To characterize the TetR gene regulation landscape, we utilize a fluorescence-based in vivo method known as sort-seq[28,32,64] to map thousands of *tetO2* variants to gene expression levels. We specifically measure levels of GFP fluorescence intensities, which serve as a proxy for GFP expression levels[65]. Each intensity level is used to compute the ability of a *tetO2* variant to repress gene expression–its repression strength. Whenever strong repression is associated with high fitness, the resulting regulatory landscape becomes a fitness landscape. We model the evolutionary dynamics of populations subject to Darwinian evolution on such a landscape. However, we emphasize that we do not measure fitness. Instead, we measure repression strength and model adaptive evolution under the assumption that repression strength can be a proxy for fitness.

Existing evidence suggests that strong repression can indeed be associated with high bacterial fitness in the TetR system[66–68]. For example, in the natural TetR system, strong repression of the efflux pump gene *tetA* by TetR is crucial for maintaining high fitness in the absence of antibiotics. Even low levels of TetA expression can impose a substantial fitness cost under these conditions[66–68], and most mutations in the *tetO2* binding site reduce its affinity for TetR, and thus also fitness. However, environmental changes often modulate mutational effects in gene regulation[67,69–71]. It has been observed for the TetR system that strong repression is not beneficial in environments with constant presence of the antibiotic tetracycline or with continuous shifts between the presence and absence of this antibiotic[67]. Moreover, the relationship between genotypes and fitness is not necessarily monotonic[72]. That is, the strongest regulation is not always most beneficial[72–74]. The relationship between repression strength and fitness is most likely to be strong and monotonic in environments where the antibiotic may be present but only rarely so[67]. Our evolutionary analysis of the TetR regulatory landscape should be interpreted with these caveats in mind.

We study the topography of the TetR regulatory landscape and how accessible its peaks of strong repression are to adaptive evolution. We also simulate adaptive evolution on this landscape to determine whether evolving populations can easily find highly active binding sites. Our observations reveal that the landscape is highly rugged. Such rugged landscapes are thought to impede navigability[4,11,75]. However, we find that the highest peaks of this landscape–the most strongly repressing TetR binding sites–can be reached by a substantial proportion of evolving populations. While this has been demonstrated in simpler theoretical models[76,77], it has not been shown in landscapes of realistic complexity[78].

## Results

### Experimental design

To map the regulatory landscape of TetR in vivo, we engineered a plasmid-based system[79] for sort-seq experiments (Fig. 1a, b,

Supplementary Figs. S1 and S2, Methods). This system allowed us to measure the binding of TetR and the resulting transcriptional repression for each genotype in a library of TetR TFBS variants, using fluorescence as a readout in a flow cytometry assay. The higher the affinity between TetR and its binding site, the lower the resulting GFP expression. To explore the TetR adaptive landscape, we randomized eight symmetrically spaced base pair positions that are especially important for the binding of the *tetO2* sequence[54,62,80]. This resulted in a library size of 65,536 unique sequences (Fig. 1c, Methods).

### Sort-seq allows the high-throughput quantification of regulation driven by thousands of *tetO2* variants

To assess our plasmid system's ability to capture differences in repression levels of TetR binding site variants, we first measured the GFP expression driven by the wild-type sequence and four previously characterized variants cloned into our plasmid. These measurements showed expression levels consistent with previous reports[80] (see Supplementary Fig. S3).

For subsequent experiments, we used our plasmid system with the wild-type binding site as a positive control for strong transcriptional repression and a negative control without a GFP promoter to set the lower bound of fluorescence as the basal autofluorescence of bacterial cells (see Methods and Supplementary Fig. S4). We then proceeded to analyze the fluorescence of our library.

In the absence of the inducer (anhydrotetracycline, Atc), the wild type TFBS promotes strong repression (low fluorescence) (Fig. 2a, left panel), whereas the entire TFBS library shows fluorescence that varies broadly (Fig. 2a, right panel). This is the expected behavior if some variants lead to strong repression but others only lead to weak repression. In the presence of the inducer, both the wild type and the entire library experience strong de-represssion (increased fluorescence), which reflects the expected dissociation of TetR from DNA.

We sorted cells harboring our library into 13 "bins" based on their fluorescence, and deep-sequenced TFBS variants from each bin (see Methods, Supplementary Figs. S5 and S6). The resulting library comprised 48,937 genotypes, covering 75% of the $4^8 = 65,536$ genotypes within the studied genotype space (Supplementary Fig. S7). For all further analyses, we additionally applied a stringent sequencing depth threshold. That is, we analyzed only TFBS variants with at least 30 sequencing reads, which reduced our library size further to 17,851 genotypes (27% of $4^8$; Methods, Supplementary Fig. S7).

Partly because of our stringent quality filtering, the correlation of read counts for each variant was highly consistent across replicates (Pearson's correlation coefficient $R = 0.971–0.991$; Supplementary Fig. S9), indicating high technical reproducibility. We then used the observed distribution of individual variants among bins to map genotypes to their respective repression levels (see Methods, Supplementary Figs. S8 and S10). We normalized repression levels by the wild type, such that values below one indicated weaker repression than the wild type, while values above one indicated stronger repression. We observed a broad distribution of repression strengths that is slightly skewed towards low values ($0.26 \pm 0.56$, mean ± s.d., Fig. 2b), indicating that most mutations generate low-repression genotypes.

Fluorescence-based sorting methods are inherently noisy, especially at the highest and lowest bounds of the fluorescence distribution[64,81,82]. To assess how such variability could impact our binding strength estimates, we measured the correlation of our repression strength measurements between replicates. We found that the Pearson's correlation coefficients for replicates were high, ranging from 0.84 to 0.92 (Supplementary Fig. S11). To validate our regulatory strength metric further, we compared fluorescence levels from a plate reader assay with our sort-seq calculated repression levels for 15 variants from each bin, for a total of 195 variants. The correlation between the two measurements was strong and nearly linear (Pearson $R = -0.86$, $p < 0.001$, $N = 195$) (Methods and Fig. 2c). To further validate

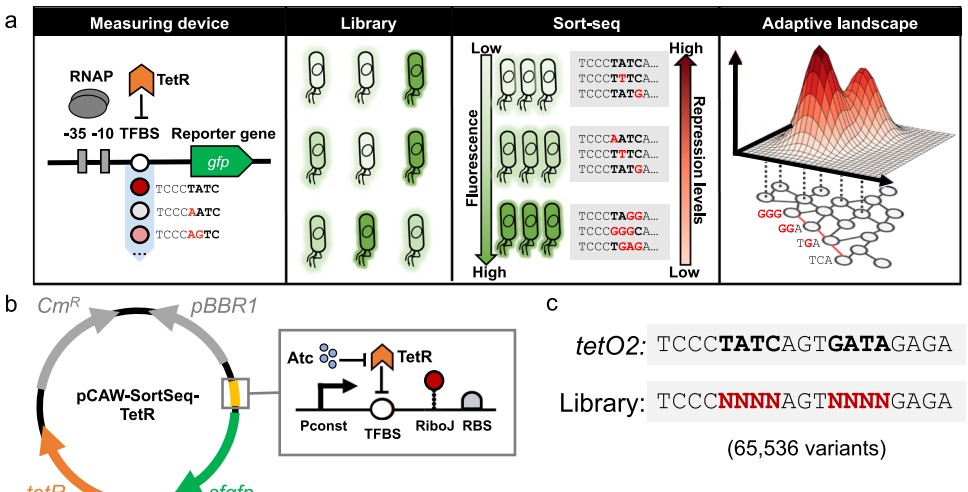

**Fig. 1 | Experimental workflow. a** Sort-seq allows the mapping of gene regulatory landscapes. We developed a modular genetic system to quantify the strength of transcriptional repression in prokaryotes. The system relies on steric hindrance, with the TFBS (or a library thereof) placed between the core promoter (−35/−10) and the *gfp* gene. When the repressor binds the TFBS, it physically blocks the RNA polymerase activity, reducing GFP expression. A TFBS library creates a population with varying GFP levels, which we sorted into 13 bins using fluorescence-activated cell sorting (FACS). After extracting, barcoding, and sequencing the genotypes, we mapped sequences to repression levels based on the sequence counts among different bins. We then built a network of genotypes (grey circles) in which sequences that differ in a single base pair are connected by an edge. The red edges in the network highlight a group of genotypes connected by single substitutions. This network, together with the repression level conveyed by each sequence, constitutes the adaptive landscape we analyze. **b** The plasmid system. The plasmid encodes a chloramphenicol resistance gene (Cm^R, grey, right-oriented arrow) and a low-copy replication origin (pBBR1, grey, left-oriented arrow). The first module contains the constitutively expressed *tetR* gene (orange), whose product represses transcription in the second module (yellow and green). TetR binds its TFBS (or a library variant) in the regulatory region (yellow, magnified in the grey box) and represses transcription (blunt vertical arrow). The inducer anhydrotetracycline (Atc, blue) inhibits TetR (blunt horizontal arrow), promoting the transcription of *gfp* (green). Transcriptional effects from the TFBS sequences on the mRNA of *gfp* are insulated by the RiboJ element (red). The ribosome binding site (RBS) for the *gfp* gene is also shown (grey). **c** Library overview. We designed the *tetO2*-derived library by randomizing eight symmetrically-spaced positions of the *tetO2* binding site (black bold letters) that are known to be important for TetR recognition[62,80]. The randomized sites (red bold Ns) are two symmetric palindromes of 4 base pairs each, yielding $4^8$= 65,536 library sequences.

the regulatory strength differences observed in the lowest bins (TFBSs with the highest repression strengths), we measured the expression levels of 45 sequences from these bins using a plate reader and compared them to the WT. Our results showed that the selected sequences exhibited significantly higher repression strength than the WT, with a relative repression increase of 14.6% ± 3.2% (mean ± s.d.). This difference was statistically significant (Welch one-sample *t* test, $t = -55.737$, df = 44, *p* value < $2.2 \times 10^{-16}$; see Supplementary Fig. S12).

In our library, 78 variants showed stronger repression than the wild type, and we analyzed these variants to characterize the consensus of strong binding sites. We created frequency matrices for each nucleotide position based on an alignment of these variants. Displayed both as a heat map and sequence logo (Fig. 2d), this frequency matrix reveals conserved nucleotides similar to the wild type (Fig. 2d, grey) and additional nucleotides at each position that increased repression relative to the wild type. Our in vivo observations are consistent with previous in vitro data (Supplementary Fig. S13).

**The TetR binding site adaptive landscape has multiple peaks**

We used a network representation to study the adaptive landscape of the TetR binding sites (Methods). In this representation, each node corresponds to a binding site variant (genotype) with its respective level of repression. Variants separated by just one nucleotide (direct neighbors) are linked by a connection (edge) denoting a single mutation. Within this network, an evolutionary path consists of a chain of consecutive mutations. We found that the vast majority of the variants we examined (17,765, 99.5%) are part of the largest connected subgraph, also known as the "giant component"[83]. This particular subgraph represents the adaptive landscape we studied further (Supplementary Table S1).

A principal indicator of an adaptive landscape's ruggedness is its number of peaks[4,84]. In our landscape, a peak is a genotype *g* whose neighbors all have a lower repression score than *g* itself (Methods). We found that the landscape has 2092 peaks (12 percent of the total number of genotypes) and is thus highly rugged. The vast majority of these peaks (2034/2092, 97%) convey repression weaker than the wild type (Fig. 3a). Only 58 peaks conveyed stronger repression than the wild type (Fig. 3a). We refer to all 58 peaks with scores above *tetO2* as strong (repression) peaks or simply as high peaks.

Because we had filtered our sequence data for quality, we had obtained repression strength data for only some (~27% of $4^8$) genotypes before mapping the landscape. This partial sampling may have introduced systematic biases in landscape topography, particularly in peak assignments[85,86]. However, several analyses suggest that such sampling biases, if they exist, may not be strong. First, the sampled sequences are not strongly biased by repression strength. Specifically, we analyzed the relative connectivity of genotypes, defined as the fraction of each genotype's 24 possible neighbors for which we have regulation data (average relative connectivity: 0.34 ± 0.17 per genotype, mean±s.d.) This analysis revealed only a weak correlation between connectivity and repression strength ($R = -0.05$, *p* value = $2 \times 10^{-14}$, Supplementary Fig. S14), indicating that strongly and weakly repressing genotypes are represented with approximately equal frequency. Furthermore, the relative connectivity of peak genotypes is also only weakly correlated with repression strength ($R = -0.05$, *p* value = $2 \times 10^{-14}$, Supplementary Fig. S14). Unrelatedly, we note that the high prevalence of sign epistasis in our landscape (Supplementary Table S1) further supports the notion of a highly rugged landscape. In sum, these analyses suggest that our landscape sampling did not have a strong impact on the aspects of landscape topography we study.

Finally, comparing our observed number of peaks (2092) with the expected number for a landscape with randomly assigned repression

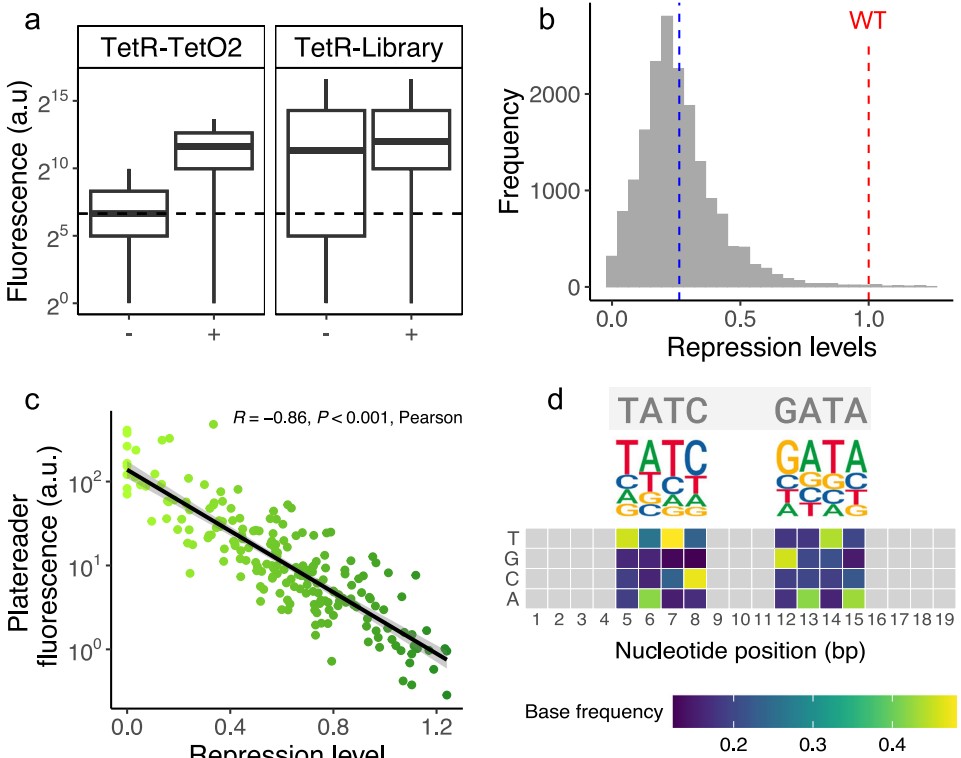

**Fig. 2 | The measuring system captures repression by TetR binding sites. .**
**a** Distribution of fluorescence levels between control and library. Each box shows the interquartile range (IQR) with the median marked by a horizontal line and whiskers extending 1.5 times the IQR. The dashed line represents the auto-fluorescence from the negative control (promoterless GFP). The positive control (left panel, the *tetO2* binding site) strongly represses GFP expression in the absence of the inducer Atc (−). This repression is alleviated (fluorescence increases) in the presence of inducer (+). The same holds for the TFBS library (right panel), except that in the absence of the inducer, most TFBSs bind the repressor more weakly than the wild type (higher fluorescence). **b** Distribution of repression levels (relative to *tetO2*) in the library. The dashed blue line indicates the mean repression of all library variants (0.26, $N = 17,851$), and the red dashed line indicates the repression level of the wild type. A total of 78 variants (to the right of the red dashed line) repress expression more strongly than the wild type. **c** Plate reader validation of

repression levels. We measured the fluorescence of 195 variants (5 variants × 13 bins) in a plate reader. We correlated their fluorescence levels (horizontal axis, arbitrary units, logarithmic scale) with our calculated repression levels (vertical axis). *R* indicates the Pearson correlation coefficient ($R = −0.86$, two-sided Pearson's correlation *t* test: $P < 0.001$). The grey shade around the regression line represents one standard error of the estimate at a confidence level of 95%. **d** Sequence logo for strong TetR binding sites. Grey letters on top indicate the wild-type (*tetO2*) nucleotides. We aligned the 78 sequences with repression levels higher than the wild type to construct a DNA sequence logo for strong TetR TFBSs. Below, a heatmap shows the frequency matrix for each variable position, with the vertical axis displaying each base that can be found at each sequence position (horizontal axis). The heatmap color gradient represents the frequency of each base at each position (see color legend). Grey cells represent TFBS positions that were not mutated in the library.

levels reinforces our assessment of high ruggedness. To estimate the number of peaks in uncorrelated landscapes, we randomly shuffled the repression strength distribution of our experimentally mapped landscape 1000 times among the landscape's genotypes. For each of these 1000 randomized landscapes we then assessed the distribution of the number of peaks (Methods). This resulted in a normal distribution with an average of 2721 ± 92 peaks (mean ± s.d.), and an upper bound of 2872 peaks at a 95% confidence level (the upper bound at this confidence level is the value below which 95% of the observations fall, as described in Methods). Thus, the number of peaks in the biological landscape is only 24% percent lower than that of randomized landscapes.

Next, we investigated the spatial arrangement of the 2092 peaks in the biological landscape. For this purpose, we examined if different peaks lie close to each other by quantifying their distribution of pairwise genetic distance—the smallest number of mutations needed to transform one peak genotype into another. For comparison, we also computed this same distance distribution for pairs of 2092 non-peak variants selected at random from the landscape. The mean distances are almost identical ($d = 5.59$ for peaks vs. $d = 5.5$ for random variants, ($d = 5.59$ for peaks vs. $d = 5.5$ for random variants, two-sided Kolmogorov–Smirnov test $D = 0.03$, $p < 2.2 × 10^{-16}$, $N_1 = 2,187,186$,

$N_2 = 2,187,186$, Fig. 3b). This pattern of similarity, however, does not apply to the high repression peaks. They are much closer together in sequence space ($d = 3.86$ for high peaks vs. $d = 5.46$ for random variants, two-sided Mann–Whitney $U = 412,662$, $p < 2.2 × 10^{-16}$, $N_1 = 1653$, $N_2 = 1653$, Fig. 3c). However, their average distance of almost four mutational steps implies that high peaks are also not confined to a small region of genotype space.

The wide scattering of peaks can also be observed in Fig. 3d, which displays the result of a principal component analysis (Methods, Supplementary Figs. S15 and S16). The PCA reveals two "clouds" of genotypes (Fig. 3d), whose members are distinguished by the nucleotide at position 12 (Supplementary Fig. S15). Genotypes with a G at position 12 fall into one cloud, whereas genotypes with A, C, or T at this position largely fall into the other cloud (Supplementary Fig. S15). We observed that 75% of peaks and 63% of high peaks fall into this second cloud (Fig. 3d). In sum, the TetR binding site landscape features numerous and widely spread adaptive peaks, with high repression peaks being more similar than low repression peaks and randomly chosen variants.

## High peaks are moderately accessible
A large population evolving by mutation and natural selection cannot traverse the valleys that lie between the high peaks of a rugged

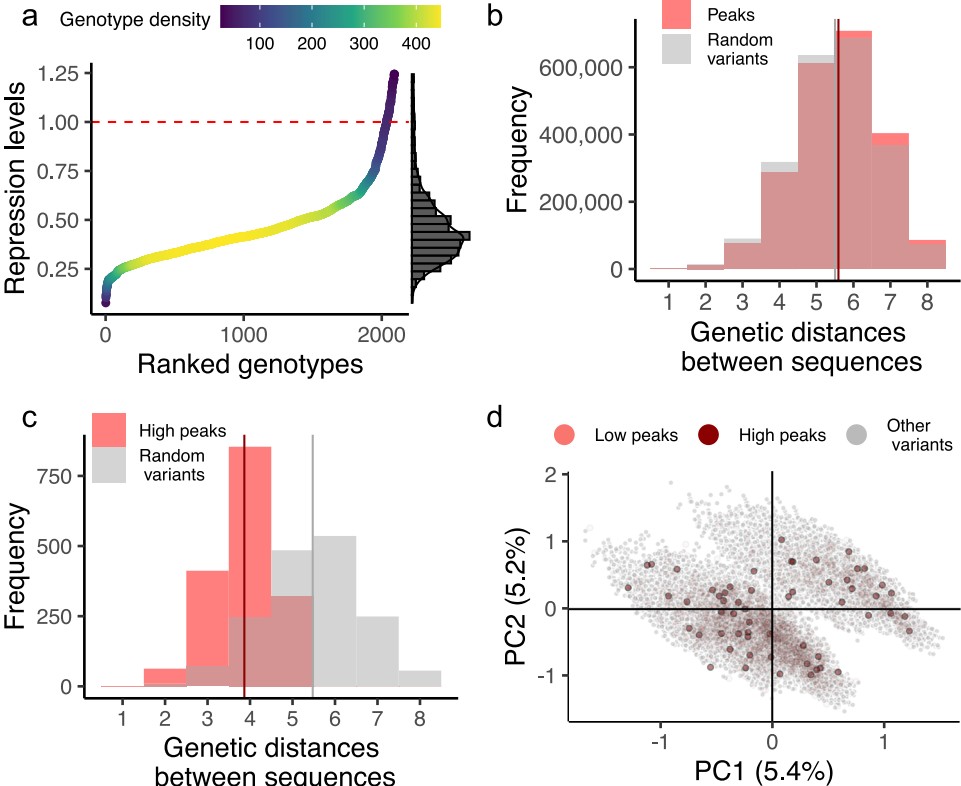

**Fig. 3 | The TetR landscape has multiple adaptive peaks with widely varying repression strengths.** . **a** Distribution of repression levels for 2092 peaks in the landscape. The dashed red line at $y = 1$ shows the repression level of the wild type. Gradient colors (see color legend) indicate the density of genotypes clustered along the horizontal axis. The histogram on the right represents the distribution of repression levels. **b** Peaks are genetically diverse. Pairwise genetic distances between nucleotide sequences are shown for 2092 peaks (red) and 2092 random non-peak variants (grey). The vertical lines represent the mean values for these distributions ($d = 5.59$ for peaks [red] vs. $d = 5.5$ for random variants [grey]). **c** High repression peaks are genetically less diverse than other variants. The figure compares genetic diversity between the nucleotide sequences of 58 high peaks (red) and 58 randomly selected non-peak variants (grey). The mean values for these distributions are indicated by vertical lines ($d = 3.86$ for the dominant peaks [red] against $d = 5.46$ for the random variants [grey]). **d** A principal component analysis reveals that adaptive peaks are distributed widely within the genotype space (Methods). The panel shows principal components (PC) 1 and 2 and the amount of variation (in percentages) explained by each component. Every circle corresponds to one among the 17,765 variants within the landscape. Dark red and light red circles represent high and low peaks, respectively, while grey circles represent non-peak variants.

adaptive landscape, because it will get stuck along evolutionarily inaccessible evolutionary paths to a high peak[87,88], i.e., paths on which repression strength decreases at least for some mutational steps. Such paths are inaccessible, at least in large populations like that of *E.coli* (with effective population size $N \approx 10^8$)[89], where even weakly deleterious mutations are unlikely to go to fixation[52]. Because the TetR landscape we study is rugged, the gradual evolution of strong TetR binding sites from weak ones might face this obstacle.

To find out whether this is the case, we studied the evolutionary accessibility of high peaks. To this end, we first determined for each binding site variant whether paths to each high repression peak exist that are evolutionarily accessible, i.e., paths in which each mutational step increases the repression strength of an evolving TetR binding site. We quantified the size of each peak's basin of attraction, i.e., the number of variants from which the peak is accessible (Fig. 4a). We found that the basin sizes of high peaks vary widely, ranging from peaks accessible from only a single variant (1/15,673) to others being accessible from 52.5% (8231/15,673) of all variants (Supplementary Fig. S17). Basin sizes also vary considerably with peak repression strength (Fig. 4b). They comprise, on average, $4.5 \pm 6$ (mean ±s.d.) percent of variants (946/15,673), but low peaks generally have basins with significantly smaller size (median: 276, 1.7 % of all variants) than high peaks (median: 1260, 8%; two-sided Mann−Whitney $U = 90680$, $p = 2.81 \times 10^{-12}$, $N_1 = 58$ $N_2 = 2034$) (Fig. 4b). Moreover, when considering all peaks, we observed that peak genotypes with higher repression levels also have larger basins of attraction (Pearson $R = 0.43$, $p < 0.001$, $N = 2092$; Fig. 4c).

Given that high-repression peaks possess large basins of attraction, one would expect that individual variants belong to multiple basins, enabling adaptive evolution to reach several repression peaks from them. This is indeed the case for 57.2% (8965/15,673) of variants (Fig. 4d).

We then quantified how many variants are shared between the basins of attraction for a pair of high peaks, and did so for all pairs of high peaks (Fig. 4e). This basin overlap varies widely. Some pairs of high peaks share 80% of the variants in their basins, while others do not share any variants. On average, the basins of high peaks share $17\% \pm 14\%$ (mean ±s.d.) of the variants in their basins (Fig. 4f). Although the sharing of basin members varied between peaks, the basins of high peaks share more variants than those of low peaks (two-sided Mann−Whitney $U = 2,633,301,522$, $p < 2.2 \times 10^{-16}$, $N_1 = 1653$, $N_2 = 2,067,561$). In sum, high peaks are more accessible than low peaks and share a greater proportion of their basins of attraction.

In a perfectly smooth landscape, the shortest accessible path between a variant and a peak equals the genetic distance between the two. However, within a multipeaked landscape like ours, accessible routes may be substantially longer than the genetic distance. In the regulatory landscape of TetR, the shortest accessible paths that terminate at a given high peak from any one variant are on average two

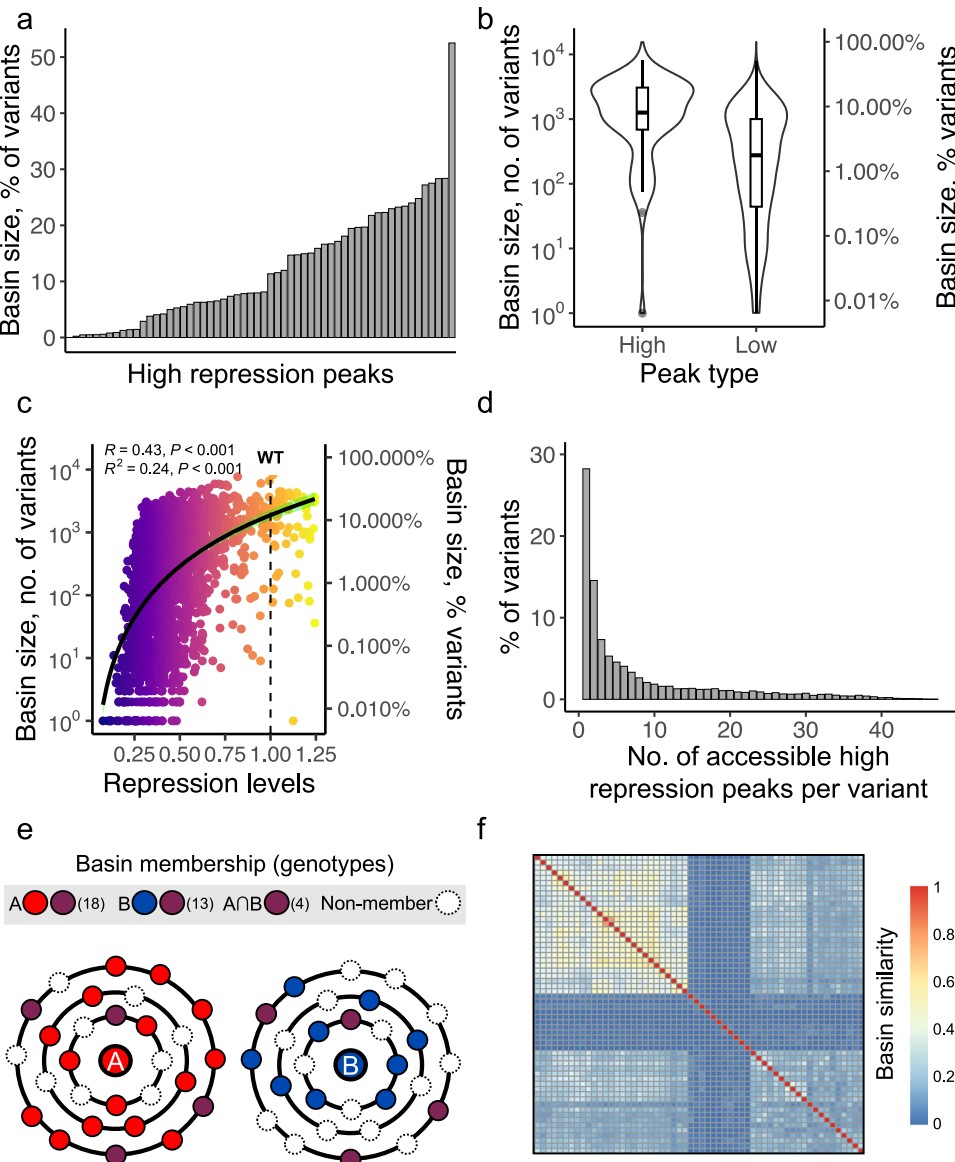

**Fig. 4 | Basins of attraction. a** The basin sizes of high peaks. Each peak's basin size (*y* axis) is shown as the percentage of variants from which the peak can be accessed. **b** High peaks have larger basin sizes than low peaks. The left y axis shows basin sizes as the number of variants from which each peak can be accessed, and the right *y* axis by the proportion of such variants (logarithmic scale). Data are split into high peaks (*N* = 58) and low peaks (*N* = 2034). Violin plots show basin size distributions. Each box shows the IQR with the median marked by a horizontal line and whiskers extending 1.5 times the IQR. **c** High peaks tend to have larger basins of attraction. The scatter plot shows the association between repression levels of peaks (*x* axis) and the size of their basins of attraction (absolute numbers on the left *y* axis, percentages on the right *y*-axis, logarithmic scale). Heatmap colors indicate repression levels from low (dark blue) to high (yellow), with the dashed line marking the wild-type level. The Pearson correlation is *R* = 0.43 (*N* = 2092, two-sided Pearson's correlation *t* test: *P* < 0.001), and the black curve represents the linear regression (*R*² = 0.24, two-sided linear regression *t*-test: P<0.001), with a 95% confidence interval shaded in green. **d** Variants can attain more than a single peak. The histogram shows the distribution of the number of high repression peaks that are accessible from each non-peak variant. **e** Schematic of basin size and overlaps between basins of attraction. Each circle represents a hypothetical genotype. Peaks A (red) and B (blue) are surrounded by rings showing genotypes at increasing mutational distances. Red, blue, purple, and white circles indicate genotypes in the basin of only A, only B, both, or neither, respectively. The number of genotypes for A, B, and their intersection are shown in parentheses. **f** The basins of attraction associated with several high peaks overlap modestly. The heatmap shows the fraction of genotypes shared across basins of attraction for 58 high peaks. Each matrix entry represents the overlap (Jaccard index) between peak pairs, with red (1) indicating identical basins and blue (0) indicating no shared variants. Rows and columns are clustered by complete linkage hierarchical clustering[165].

mutations longer (7.53 ± 2.63 mutations, mean±s.d.) than the mean genetic distance between the variant and the peak (5.43 ± 1.33 mutations, mean±s.d.) (Fig. 5a, Welch two sample *t* test, *t* = −289.92, df = 247078, *p* value < 2.2 × 10⁻¹⁶, *N* = 166,417).

Thus far, we have defined a peak as accessible from any one variant if monotonically repression-increasing paths to it exist from the variant. However, the fraction of such accessible paths among all possible paths may be tiny, such that evolving populations may usually not find them. To find out whether this is the case, we enumerated all

paths, both accessible and inaccessible, from every non-peak genotype to every peak genotype (Methods). Our analysis revealed an exponential increase in the number of paths to a peak as the genetic distance between the variant and the peak increased (Fig. 5b). We note that for a combinatorially complete landscape, the number of direct paths would be expected to grow factorially with genetic distance (d!), i.e., there are 40,320 paths for genotypes separated by a genetic distance 8 mutations. However, due to the incompleteness of our landscape, the number of paths we observe is lower (Fig. 5b). At short

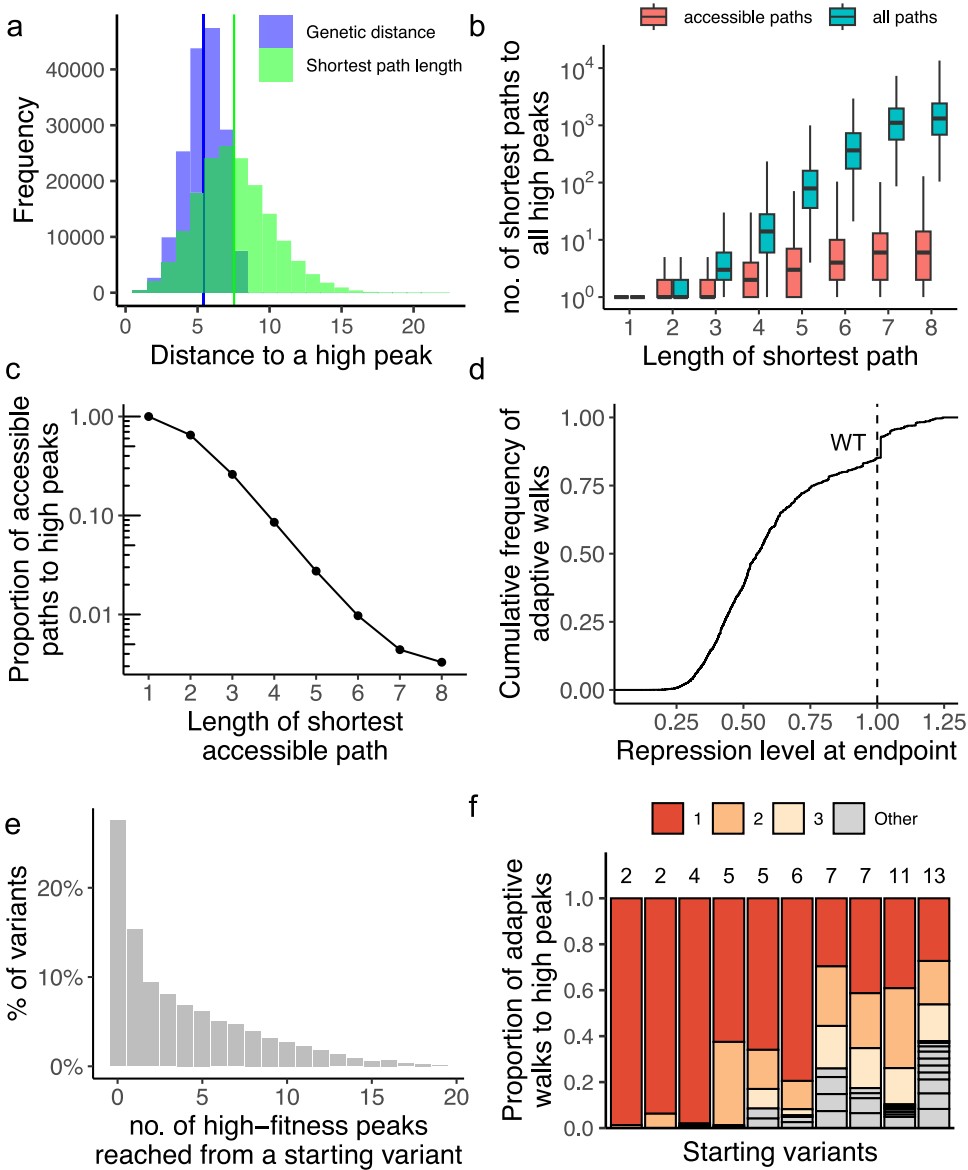

**Fig. 5 | Peak accessibility. a** Accessible paths to high repression peaks tend to be short. The blue histogram shows the distribution of the genetic distances between all pairs of variants and their accessible peaks. The green histogram shows the distribution of the number of mutational steps in the shortest accessible paths between these variants and their accessible peaks. **b** The landscape harbors many paths leading to high repression peaks. The vertical axis shows the number of shortest paths per variant to a high peak, while the horizontal axis shows the length of these shortest paths. Red and blue boxplots summarize all shortest paths and accessible shortest paths, respectively. Each box shows the IQR with the median marked by a horizontal line and whiskers extending 1.5 times the IQR. **c** The fraction of accessible paths decreases as path length increases. The vertical axis (logarithmic scale) shows the fraction of accessible paths among all shortest paths to high repression peaks. Circles indicate the average fraction for each path length **d** High repression peaks are attainable through adaptive evolution. The panel shows the

cumulative distribution function of repression values reached by $10^3$ adaptive walks starting from each non-peak variant. The dashed vertical line at $x=1$ shows the wild-type repression level, with only 20% of walks reaching or exceeding this level. **e** Most starting variants can reach more than a single peak. The bar plot shows the proportion of variants reaching zero, one, or multiple high peaks. We simulated $10^3$ adaptive walks for each non-peak variant ($N = 15,671$). Random walks starting from 27%, 15%, and 58% of variants reached no peaks ($x = 0$), exactly one peak ($x = 1$), or more than one peak ($x > 1$), respectively. **f** Some high repression peaks are reached more frequently than others. We randomly selected 10 starting variants, each represented by a bar, with stacks showing the number of high peaks reached during $10^3$ adaptive walks. The numbers above the bars indicate the total number of peaks reached. Stacks represent the proportion of walks reaching each peak (totaling 100%). Stacks are ordered by frequency, with the top three peaks color-coded in red, light orange, and light yellow.

distances from a peak, a large fraction of paths is accessible, but this fraction decreases dramatically as the path length increases (Fig. 5c). At the maximum distance of eight mutational steps, merely 1% of paths remain accessible.

## Only a minority of evolving populations reach high repression peaks

The length of accessible paths to high peaks (Fig. 5a), their smaller numbers (Fig. 5b, c), and the rarity of high repression peaks (only 3% of

all peaks), raise the question of whether evolving populations would ever discover these peaks. To find out, we simulated evolution on the TetR landscape by random mutation and natural selection for strong TetR-mediated repression.

Because *E. coli* has large populations with more than $10^8$ individuals[89], even small effects of mutations are visible to natural selection. In addition, the probability of mutations occurring in our 8 nucleotide positions of the wild-type binding site in the *E. coli* genome is very low (8 positions × $2.2 \times 10^{-10}$ mutations per position per

generation[89]). Thus, adaptive evolution on our landscape would occur in the well-studied strong selection weak mutation regime (SSWM)[40,90–92]. In this regime, any beneficial mutation is likely to become fixed before another mutant appears that will eventually go to fixation. In other words, populations are monomorphic most of the time–they occupy a single genotypic position on the landscape–and undergo an uphill walk until a peak is reached. In this regime, clonal interference between selected variants and recombination can be neglected[92,93].

We simulated such adaptive walks as starting from each genotype in the landscape that is not a peak. We then chose at random and with uniform probability one of this genotype's (1-mutant) neighbors with higher repression than itself as the first step in this random walk. We continued this procedure for as many steps as needed until a peak had been reached. More specifically, we performed $10^3$ simulations for each of the 15,671 non-peak starting genotypes.

Adaptive walks that terminated on high repression peaks comprised on average $6.2 \pm 2.8$ (mean±s.d.) mutational steps, 2.6 mutations more than the mean shortest genetic distance between the corresponding start and end points of the adaptive walks ($3.6 \pm 1.5$, mean ±s.d., Supplementary Fig. S18a., Welch Two Sample $t$-test, $t = -1259.8$, df = 3603271, $p$ value $< 2.2 \times 10^{-16}$, $N = 2,353,470$). Notably, most walks reached low repression peaks, with only 20% terminating at high peaks (Fig. 5d). This number is modest, but higher than expected when considering that only 3% of all peaks are high peaks. We note that the discontinuity between $y=0.8$ and $y=0.9$ observed in Fig. 5d results from the distribution of peak repression strengths at high repression strengths, where peaks become increasingly rare (Fig. 2b)

Next, we wanted to find out whether population size affects this observation. In small populations, genetic drift can help populations traverse the valleys between a low peak and a higher peak, raising the question of whether small populations may attain higher peaks[1,94]. We simulated adaptive walks with an approach pioneered by Kimura[88,95] that allows us to calculate the fixation probability of any mutation as a function of population size and the selection coefficient $s$, i.e., the difference in repression strength between neighboring genotypes[88,96]. We used this approach to simulate adaptive evolution in populations with $10^8$, $10^5$ and $10^2$ individuals, and found that 20%, 25%, 20% of walks attained high peaks, respectively (Supplementary Fig. S19).

Finally, we also studied how deviations from the SSWM regime would affect our observations. Populations outside this regime are polymorphic most of the time[93,97,98] and as a result, they experience clonal interference, in which the fittest of several simultaneously segregating mutations is likely to go to fixation[93,97,98]. We simulated this scenario through "greedy" adaptive walks, in which an evolving genotype steps to the neighbor with the highest repression increase until a repression peak is reached[11,91,99,100]. Because such walks are deterministic when started from the same genotype, we performed only one simulation for each non-peak genotype ($N = 15,671$). Similar to walks in the SSWM regime (Supplementary Fig. S20f), 20% percent of populations attained a high repression peak. In sum, regardless of the regime we studied, a substantial minority of evolving populations reached a high repression peak.

Lastly, because gene expression and sort-seq experiments are noisy[64,81,82], we asked how such noise could influence landscape topography and adaptive evolution on the landscape[4,101,102]. To this end, we estimated an individual experimental noise value ($\tau$) for the repression strength $S_A$ and $S_B$ for each focal genotype A, and for each of its single mutants B, based on the standard deviation of their repression strengths across our three replicate experiments (Methods). We then considered the repression strength $S_B$ of a mutant to be indistinguishable from $S_A$ if its average across the replicates lies within the experimental noise of $S_A$ (Methods). We mapped the regulatory landscape under this assumption for varying values of this noise (Methods). Not surprisingly, as noise increases, the landscape becomes smoother,

the number of peaks decreases, and peaks merge into increasingly broader plateaus (Supplementary Fig. S21). Accessibility of these peaks also increases (Supplementary Fig. S21), which is reflected by an increasing proportion (25% to 37%) of adaptive walks that reach at least one high repression peak (Supplementary Fig. S22).

In addition, we observed that the number of times a genotype is visited during an adaptive walk is not highly sensitive to noise (Supplementary Fig. S23). This is relevant because it shows that regardless of landscape changes caused by noise, adaptive walks traverse the same genotypes with similar frequency, indicating similar evolutionary dynamics.

## Adaptive evolution on the TetR landscape is highly contingent on chance events

Because different peaks show overlapping basins of attraction (Fig. 4f), adaptive evolution on the TetR landscape may be contingent on the starting genotype and on the mutational steps that a population takes. Both factors may influence which peak the population reaches. More generally, evolutionary contingency refers to the dependence of a historical process on chance events that can render this process unpredictable[103,104]. To quantify the extent of such contingency, we first simulated 1000 SSWM adaptive walks starting from each non-peak genotype, assessing the number of different high peaks that can be reached from each such genotype .

Among starting genotypes from which adaptive walks attained at least one high peak, 58% reached two or more peaks (Fig. 5e). These observations agree with our earlier observation that multiple high repression peaks are accessible from individual genotypes (Fig. 4d). Not all peaks are reached with the same probability, i.e., some are reached more frequently than others (Fig. 5f). Finally, we note that many alternative shortest paths can be accessed from a starting genotype to a peak ($6.37 \pm 8.43$ mean±s.d, Fig. 5c). In sum, our observations suggest that the identity of any peak attained on the TetR landscape is highly contingent on stochastic events during adaptive evolution.

## Discussion

Understanding the interplay between the spaces of the "actual"–what already exists in nature–and the "possible"–what could exist–is a fundamental endeavor in evolutionary biology[105–107]. Here, we address this problem by exploring how evolution can navigate a space of possible genotypes embodied in a library of more than 10,000 TFBSs. We did so for the prokaryotic TF TetR[52,57], a transcriptional repressor with only two known natural cognate binding sequences and fewer than a dozen synthetic TFBSs[58–60,80]. To this end, we developed a plasmid-based system to characterize more TetR binding sites and study the adaptive landscape formed by these variants. Our experiments expand the available information on TetR TFBSs from less than a dozen to thousands of mutational variants.

We found that 12% of the variants in our landscape are peaks. By this measure, the TetR landscape is highly rugged. It is also rugged by another measure, a non-additive interaction between mutations known as reciprocal sign epistasis, in which two individual mutations reduce repression strength but both mutations together increase it (Methods)[108–110]. Specifically, 30% of mutant pairs in our landscape exhibit reciprocal sign epistasis (Supplementary Fig. S24, Methods). This is a much higher incidence than in smooth and better-studied eukaryotic regulatory landscapes, where only 4,5% of mutant interactions in TFBSs exhibit reciprocal sign epistasis[4]. Well-studied protein adaptive landscapes also show less reciprocal sign epistasis (8–22%[102,111–113]). However, recent studies of epistatic interactions in TFBSs of bacterial and phage TFs (LacI[108,114], AraC[69] and CI[71]) have shown that sign epistasis is pervasive in these TFBSs. It occurs in more than half of the studied mutants.[69,71,108,114]

In our TFBS library, only a tiny fraction (0.44%) of peaks conveys higher repression levels than the wild type. Despite the rarity of strong

repression peaks and the high ruggedness of this landscape, it also has properties typically associated with smooth landscapes. Specifically, we observed that higher repression peaks are more accessible, in the sense that they have larger basins of attractions, than lower peaks. In addition, 20% or more of evolving populations reach peaks, conveying higher repression than the wild type. This holds for both large and small populations and in the presence of clonal interference (Methods). It suggests that indicators such as the number of peaks may be misleading when studying the evolutionary accessibility of high peaks in rugged landscapes. It is also consistent with previous work showing that the relationship between peak accessibility and epistasis in complex landscapes may not be straightforward[113,115,116].

We considered all mutations in our analysis as non-neutral with respect to repression levels. When assuming extensive neutrality by simulating adaptive walks with low population sizes ($N = 10^2$, Methods and Supplementary Fig. S19), the number of accessible paths to high-peaks became even higher, because paths passing through slightly deleterious mutations became accessible. This is not surprising, because drift can help populations traverse valleys in the landscape[1,94]. In other words, our estimates of peak accessibility are conservative.

One possible albeit speculative explanation for the navigability of the TetR binding site landscape is that it has been shaped by the evolution of TetR itself through the deleterious consequences of misregulated *tet* resistance genes. In nature, TetR regulates the expression of the TetA efflux pump[56,61] whose loss of regulation is toxic to *E. coli*, partly because it leads to a loss of membrane potential[57,68,117,118].

Our observation that it is easy to evolve high repression TetR binding sites raises the question of why such sites have not been observed in nature. Aside from the possibility that they remain to be discovered, there may also be a limit to which high repression may benefit TetR. For example, there may be a trade-off between the reduction of the fitness cost conferred by tight repression and the capacity of TetR to quickly respond to low concentrations of the tetracycline inducer[57,69]. Such a trade-off has been observed for the LacI repressor, which regulates the lactose-metabolizing *lac* operon. Excessive binding affinity of LacI to its TFBS requires a higher concentration of the inducer to de-repress the system[119]. If the same applied to TetR, the necessary increase of intracellular tetracycline concentration may itself be toxic. More generally, the assumption that the strongest regulation is the most beneficial may not always apply[72–74].

The evolutionary analysis of the TetR regulatory landscape is complicated by the fact that the relationship between repression strength and fitness can depend on the environment[69,70]. A previous experimental study demonstrated strong epistatic interactions between the mode of gene regulation of the tetracycline resistance *tet* operon (inducible by TetR or constitutively expressed) and the promoter strength of the resistance gene (low, medium, or high)[67]. These interactions are influenced by different tetracycline concentrations, suggesting that determining the optimal mode of expression (constitutive or inducible) requires knowledge of the promoter strength of the *tet* operon[67]. The same study also showed that inducible antibiotic gene expression may be detrimental in environments with fluctuating antibiotic presence. Notably, populations in which the *tet* operon is tetracycline-inducible experience a prolonged lag phase when transitioning from an antibiotic-free environment to one containing antibiotics[67]. Additionally, in environments with sustained antibiotic presence, the complete induction of resistance renders inducible expression less efficient than constitutive expression, because TetR inhibits full expression of the resistance operon[67]. An advantage of our approach to mapping a gene repression landscape, rather than a fitness landscape, is that it allows the study of evolution under various

relationships between repression strength and fitness. For instance, fitness might be highest at intermediate or low repression strengths.

We observed extensive contingency in the adaptive evolution of high repression genotypes, because populations starting from the same low repression genotypes often reached different high repression peaks. This genotypic contingency may impact future phenotypic evolution for two reasons. First, different high repression peaks may confer varying repression strengths and, consequently, different fitness. Second, environmental variation, such as in temperature, inducer concentrations, or protein levels, may exacerbate these differences, similar to what has been observed in other bacterial transcriptional regulation systems[69]. Previous studies with the Lac repressor[70,114] and the AraC[69] repressor demonstrated that alternating environments opened new adaptive trajectories and led to new epistatic interactions dependent on the presence of inducers. Through such environmental variation in fitness, different peaks in a landscape can become departure points for different evolutionary trajectories, further reducing the predictability of evolution[120,121].

Our work also shows that high repression peaks are more accessible, which aligns with theoretical predictions from simple theoretical models of adaptive landscapes. Specifically, in the House-of-Cards (HoC) model where neighboring genotypes have uncorrelated fitness values[11,77], peak accessibility correlates with peak height[122,123]. More generally, Das et al.[78] described a novel category of rugged yet highly navigable landscapes, which shows that empirical fitness landscapes can display properties not typically predicted by simple statistical models[84,124,125]. Schmiegelt and Krug[122] also noted that the HoC model's assumption of independent and identically distributed random fitness values does not reflect the varying degrees of fitness correlations seen in empirical landscapes. In this context, our study underscores the complexity of biological data. It also highlights the importance of empirical studies in validating theoretical models, and motivating future theoretical work that hews close to experimental data.

In sum, we introduced an assay platform to investigate bacterial regulatory landscapes in vivo, allowing us to expand previous studies from a limited number of sequences to thousands. This system also offers a versatile tool for promoter engineering, with applications in biotechnology and synthetic biology[126,127]. Our findings reveal that while the TetR landscape is highly rugged, adaptive evolution can easily reach its highest peaks. This finding adds to the increasing body of empirical evidence suggesting a non-trivial relationship between landscape ruggedness and navigability. It also highlights a need for further refinements of current landscape theory[113,115,116]. Additionally, our study paves the way for future studies comparing the landscapes of various TFs and investigating the impact of environmental changes on landscape topography and navigability.

## Methods

### Media and reagents

To prepare SOB medium, we dissolved 25.5 g of its solid stock (VWR J906) in 960 ml of distilled water and autoclaved the medium before use. To prepare SOC medium, we added 20 ml of 1 M D-glucose (Sigma G8270), and 20 ml of 1 M magnesium sulfate (Sigma 230391) to 960 ml of SOB medium. To prepare LB medium, we dissolved 25 g of its solid stock (Sigma-Aldrich L3522) in 1 liter of distilled water and autoclaved the medium before use. We purchased M9 minimal salt from Sigma (M6030), dissolved it according to the supplier's instructions, sterilized the solution by autoclaving, and supplemented it with 0.4% glucose (Sigma G8270), 0.2% casamino acid (Merk Millipore, 2240), 2 mM magnesium sulfate (Sigma 230391), and 0.1 mM calcium chloride (Sigma C7902). Where necessary, we supplemented the growth media with chloramphenicol (50 μg/mL working concentration) and/or anhydrotetracycline (100 ng/mL working concentration) (Cayman-chemicals #10009542).

## Strains and plasmids

We obtained electrocompetent *E. coli* cells (strain SIG10-MAX®) from Sigma Aldrich (CMC0004). The genotype of this strain (Supplementary Table S2) is similar to DH5α (Sigma Aldrich commercial information, see Supplementary Table S2) and is resistant to the antibiotic streptomycin. Due to its high transformation efficiency, we used it for molecular cloning, library generation, and sort-seq experiments. The design, genetic parts, and assembly of the plasmid vectors pCAW-Sort-Seq and pCAW-Sort-Seq-Neg we used in this study are available in the Supplementary material. All vectors and strains are listed in Supplementary Tables S2 and S3.

## General procedures

**Overnight incubation of cultures in liquid and solid medium.** Unless otherwise stated, we grew overnight liquid cultures on liquid LB medium (15 mL or 50 mL Falcon tubes) supplemented with chloramphenicol (50 μg/mL working concentration) for 16 hours at 37 °C and 200 rpm (50 mm orbital motion) on an Infors HT Multitron Incubator Shaker. Unless otherwise stated, we incubated bacterial colonies on solid LB-agar medium (sterile 90 mm × 15 mm plastic Petri dishes) supplemented with chloramphenicol (50 μg/mL working concentration) for 16 hours at 37 °C.

**PCR Reactions.** Unless otherwise stated, we amplified DNA fragments by PCR using a Q5® high-fidelity polymerase (NEB #M0491L) to reduce the probability of introducing mutations into the amplicons. We adopted the reaction protocol provided by NEB for a final volume of 50uL. We performed each reaction in duplicate and pooled the reaction products at the end of the PCR program. We calculated the primer melting temperatures (Tm) following the NEB Tm calculator () for a primer concentration of 500 nM.

**Verifying PCR products through gel electrophoresis.** Unless otherwise stated, after PCR amplification, we checked reactions for the presence of single-band amplicons (and the absence of unspecific bands) through electrophoresis. We performed electrophoresis in a 0.8% agarose Tris-EDTA (TAE) gel for 45 minutes at 120 V, or until the gel bands had migrated more than halfway across the total length of the gel.

**DNA purification with commercial kits.** After confirming the presence of single bands during electrophoresis, we purified the samples using the Monarch® DNA PCR/Gel Extraction Kit (NEB #T1020L), following the original protocol provided with the kit. When needed (e.g., in the presence of unspecific bands after PCR), we gel purified samples with the following protocol. We added 10 μL of 6× NEB DNA dye to each 50 μL PCR reaction, loaded the full volumes on a 1% agarose gel, and performed electrophoresis for 45 minutes at 120 V, or until the gel bands had migrated more than halfway across the total length of the gel. We used a scalpel to remove the DNA band corresponding to the amplified sequence. We performed the gel extraction using the Monarch® DNA Gel Extraction Kit (NEB #T1020L).

**DNA purification through ethanol precipitation.** We precipitated DNA by adding 1 μL of glycogen (R0551, Thermo Scientific) to 20 μL of to the inactivated ligation reaction, 50 μL of 7.5 M ammonium acetate (A2706-100ML, Sigma), 375 μL of ice-cold absolute ethanol, and 80 μL of ddH2O. After incubating at −20 °C for 20 min, we centrifuged the mixture at 18,000 × *g* for 20 min at 4 °C. We washed the precipitate twice using 800 μL of cold ethanol (70%). After drying the precipitate using an Eppendorf concentrator 5301, we dissolved it in at least 10 μL of ddH2O.

**Gibson assembly.** We performed the Gibson assembly of PCR amplified fragments with an NEBuilder-HiFi® DNA Assembly Master Mix kit (NEB #E2621L). We calculated the molarity for the assembly based on the protocol provided by the Barrick Lab (https://barricklab.org/twiki/bin/view/Lab/ProtocolsGibsonCloning). We incubated the reaction mixture for 1 hour at 50 °C in a dry bath incubator, and then placed it on ice for further use.

**Preparation of electro-competent cells.** We prepared electro-competent cells using glycerol/mannitol step centrifugation[128]. Briefly, we grew the chosen *E. coli* strains in 5 mL SOB medium at 37 °C and 250 rpm overnight. We transferred 3 mL culture into 300 mL SOB medium the next morning and continued to incubate the transferred culture at 37 °C and 250 rpm until its OD600 had reached a value between 0.4 and 0.6 (optical path length: 1 cm, 2–4 hours). We cooled the culture on ice for 15 min and collected cells at 4 °C by centrifuging at 1500 × *g* for 15 min. We used 60 mL ice-cold ddH2O to suspend the cells and distributed them into three 50 mL tubes. Then, we slowly added 10 mL ice-cold glycerol/mannitol solution (20% glycerol (w/v) and 1.5% mannitol (w/v)) to the bottom of each tube by using a 10 mL pipette. We centrifuged the tubes at 1500 × *g* and 4 °C for 15 min in a centrifuge (Eppendorf 5810/5810 R) by setting acceleration/deceleration to zero. We removed the supernatant and suspended the cells in 3.0 mL ice-cold glycerol/mannitol solution. Subsequently, we transferred 100 μL of the resulting suspensions into pre-cooled 1.5 mL tubes and incubated them in a dry ice-ethanol bath for -1 min. Then we stored the suspensions at −80 °C for transformation experiments.

**Electroporation.** For all the transformation procedures in this work, we transformed 100 μL of electrocompetent cells by electroporation in 0.2 cm cuvettes (EP202, Cell Projects, UK), through a Micropulser electroporator (Bio-Rad) set at EC3 (15k V/cm). We recovered electroporated cells in 1 mL of pre-warmed SOC media in 15 mL falcon tubes for 1.5 hours (37 °C, 220 rpm). Except when stated otherwise, we plated 300 μL of the recovered culture on an LB agar plate supplemented with 50 μg/mL of chloramphenicol and incubated overnight (16 hours) at 37 °C for subsequent screening and confirmation of clones through Sanger sequencing.

## The pCAW-Sort-Seq design

We designed the pCAW-Sort-Seq plasmid in the Snapgene® software (www.snapgene.com) by combining information from different literature sources. Below, we provide a more detailed explanation of the plasmid's components.

The pCAW-Sort-Seq plasmid (Fig. 1b, Supplementary Figs. S1 and S2) is a PSEVA231-derived plasmid[129], harbouring a pBBR1 replication origin (broad-host, low copy, i.e., 5 to 10 copies per cell)[130] and a chloramphenicol resistance gene. We chose the pBBR1 origin of replication for two main reasons. First, a low-copy system is better suited for simulating the native protein-DNA interactions observed in the single-copy bacterial chromosome. Second, our system is compatible with a wide range of gram-negative bacteria, allowing the generation of genotype-phenotype maps in different bacterial hosts. The TetR repression system we adopted in this work is based on ref. 62. We use anhydrotetracycline (Cayman-chemicals#10009542) as an inducer to derepress the system when required. We diluted anhydrotetracycline in absolute ethanol in a stock concentration (1000μ) of 100 μg/mL to a working concentration of 100 ng/mL.

The fluorescent reporter gene of the pCAW-Sort-Seq plasmid is *sfgfp*[131]. It provides information on TetR repression/binding strength. The *tetr* gene is constitutively expressed under a low-strength pLac promoter variant developed by ref. 132. Measuring TetR repression through sfGFP fluorescence relies on having a *tetO2* (5'-TCCCTAT-CAGTGATAGAGA-3') binding site between a medium strength promoter (the BBa_J23110 promoter from http://parts.igem.org/Part:BBa_J23110[133]) and the *sfgfp* gene, at the +10 position relative to the transcription starting site (TSS) of the gene[134]. The region where the *tetO2*

binding site is located can be easily replaced for generating TFBS libraries through restriction digestion (HindIII and BamHI restriction sites) or Gibson assembly[135]. The reporter gene *sfgfp* is insulated by a transcriptional insulator named RiboJ, a synthetic ribozyme that removes 5'UTR interferences from variable TFBS sequence in the mRNA by self-cleavage[136]. Promoters are preceded by insulator sequences, terminators that alleviate contextual effects of upstream sequences over promoter regions[137]. We based the divergent orientation of regulatory modules on ref. 138. We retrieved natural (*ECK*) and synthetic transcriptional terminators (*synterm*) from ref. 139. Our system is also compatible with barcoding for in vivo multiplexed measurements based on RNA-Seq, as proposed by ref. 49. We analyzed the plasmid with the EFM calculator[140] for genetic stability, and iteratively modified its design until a low instability score (RIP Score: 50.0) was reached.

### Assembling the pCAW-Sort-Seq plasmid
The designed plasmid was synthesized by Twist Biosciences (USA) in two fragments that we joined through Gibson assembly[141] using the pCAW_frag1_F/pCAW_frag1_R, pCAW_frag2_F/pCAW_frag2_R primer sets (Supplementary Table S4). For a more in-depth view of the plasmid map, see Supplementary Fig. S1, or the Addgene collection browser (https://www.addgene.org/), ID (to be uploaded).

We amplified the synthesized DNA fragments by PCR, performing each reaction in duplicate, and pooled the reaction products at the end of the PCR. We performed PCR with the following program: 98 °C/30 s; 25 cycles of 98 °C/10 s, 64 °C/30 s, 72 °C/30 s; 72 °C/2 min.

After the PCR amplification, we checked reactions for the presence of single-band amplicons and the absence of unspecific bands through gel electrophoresis. After confirming the presence of single bands, we treated the pooled PCR reactions with 2 μL of DpnI restriction enzyme (NEB #R0176L), and incubated them at 37 °C for 2 hours to remove traces of the template plasmid. We purified digested samples using the Monarch® DNA gel extraction kit (NEB #T1020L). We then performed the assembly of the two fragments with a NEBuilder-HiFi® DNA Assembly Master Mix kit (NEB #E2621L). We calculated the molarity for the assembly based on the protocol provided by the Barrick Lab (https://barricklab.org/twiki/bin/view/Lab/ProtocolsGibsonCloning).

Subsequently, we combined an equimolar ratio of the two fragments to a total volume of 5 μL, and added 5 μL of the Gibson master mix (2X). We incubated the resulting 10 μL reaction at 50 °C for 1 hour in a dry-bath incubator. For transformation, we directly added 1 μL of the reaction to 100 μL of electrocompetent cells, and performed transformation as described in the section *Electroporation*. When needed, we purified the reaction using the Monarch® DNA gel extraction kit (NEB #T1020L) prior to transformation, resuspended the purified DNA in 10 μL of nuclease-free distilled water, and used all 10 μL for transformation. After transformation, we plated recovered cultures on LB agar plates and incubated them at 37 °C overnight. We picked colonies and directly sent them for clone confirmation through Sanger sequencing (NightSeq® service, Microsynth, Switzerland). We grew individual clones overnight in LB liquid media supplemented with chloramphenicol, aliquoted them into 1 mL cryotubes as glycerol stocks (glycerol (20% v/v final concentration) and stored them at −80 °C for further use.

### Library design, synthesis and amplification
Several studies have identified the most important positions for *tetO2* DNA binding[62,80]. Based on these studies, we designed the *tetO2*-derived library by randomizing the eight most important positions. The randomized sites form two symmetric palindromes of four base pairs each. The calculated library size for this approach is $(4^8)^{-}65,536$ sequences. The originally published *tetO2*-original sequence[62] is 5'-TCCCTATCAGTGATAGAGA-3'. Our *tetO2*-derived

library sequence is 5'-TCCCNNNNAGTNNNNGAGA-3', where N represents the randomized positions.

We designed the library using the Snapgene® software (snapgene.com), and had it synthesized by IDT (Coralville, USA) as a single-stranded DNA Ultramer® of 140 bp (4nmol). We resuspended the library in nuclease-free distilled water and serially diluted it to a concentration of 50 ng/uL. We used Ultramers® as templates in a PCR reaction for the formation of dsDNA fragments and library amplification. We amplified the Ultramer® DNA molecules by PCR (Supplementary Table S5). We performed PCR with the following program: 98 °C/30 s; 25 cycles of 98 °C/10 s, 60 °C/15 s and 72 °C/80 s; and 1 cycle of 72 °C/5 min. We opted for a maximum of 25 cycles to reduce the amount of amplification biases. After amplification, we analyzed samples through gel electrophoresis to confirm that only a single band with a size -140 bp was present for each PCR reaction. After confirming the presence of single bands, we purified the samples using the Monarch® DNA gel extraction kit (NEB #T1020L). Whenever unspecific bands appeared during electrophoresis, we gel purified samples using the same kit.

### Library cloning
We digested a total mass of 1 μg of the purified library overnight (37 °C, 16 hours) by HindIII-HF (NEB #R3104) and BamHI (NEB #R3136) restriction enzymes (5 μL of each enzyme) in a reaction volume of 100 μL. We isolated the cloning plasmid using a QIAprep spin miniprep kit (Qiagen, Germany), and digested it overnight (37 °C, 16 hours) with the same restriction enzymes as the library (5 μL each) in a reaction volume of 100 μL. After the overnight digestion, we added 3 μL of Quick CIP (M0525L) to the reaction to dephosphorylate DNA ends, and to avoid self-ligation in subsequent reactions (37 °C, 2 hours). We purified samples using the Monarch® DNA gel extraction kit (NEB #T1020L).

We performed ligation following a 10:1 molar ratio of insert to vector, using 100 ng of purified vector backbone, 10 units of T4 DNA ligase NEB #M0202L, and 2 μL of 10× ligation buffer (M0202L, NEB) in a 20 μL ligation reaction. We incubated the mixture at 20–22 °C for -16 h, followed by 10 min of inactivation at 65 °C. We purified the ligation product using an in-house ethanol precipitation protocol (see section *DNA purification through ethanol precipitation*), resulting in 10 μL of resuspended DNA samples. Whenever recovery efficiency was low, we purified samples using the Monarch® DNA PCR/Gel Extraction Kit (NEB #T1020L), following the original protocol provided with the kit, and eluted the purified DNA in 15 μL of dH2O. We transformed *E. coli* cells (SIG10-MAX®) with 5 μL of the purified ligation by electroporation (see section *Electroporation*).

After transformation, we plated 50 μL of recovered cultures on LB agar plates, and incubated overnight for subsequential *cfu* (colony forming unit) counting and transformation efficiency estimation. We diluted the remaining volume of the recovered cultures (950 μL) in 9 mL of LB medium supplemented with chloramphenicol (50 μg/mL) and grew them overnight. Subsequently, we aliquoted the cultures in 1 mL cryotubes with glycerol (20% v/v final concentration) and stored them at −80 °C. Based on our *cfu* counting, we estimated the mean transformation efficiency (maximal achievable library size) within this experimental setup as $10^6$ cells per transformation for the SIG10-MAX®strain. From the agar plates, we picked 20 colonies for colony PCR followed by Sanger sequencing to preliminarily assess library diversity (NightSeq® service, Microsynth, Switzerland).

### Analysing and sorting cells
**Replicates.** Data from high-throughput methods to measure gene expression are intrinsically noisy. The reasons include biological cell-to-cell expression variation[142], as well as technical[81,143–145] variation (e.g. pipetting errors, equipment biases, limit of detection thresholds etc.). In order to account for such variability, we carried out the sorting

procedure in three replicates derived from three independent library transformations.

**Sort-seq procedure.** In preparation for cell sorting, we grew cells harboring the library in LB medium supplemented with chloramphenicol. Specifically, we separately grew 1 mL of frozen aliquots of transformed cells and a streak of cells harboring a control plasmid (promoterless pCAW-Sort-Seq plasmid, without sfGFP expression) overnight in a 50 ml Falcon tube with 9 mL of LB medium supplemented with 50 μg/mL of chloramphenicol. We then diluted the overnight cultures in LB medium supplemented with 50 μg/mL chloramphenicol in a 1:100 ratio (v/v), and grew the cultures for 5 h to late-exponential/initial-stationary phase (200RPM, 37 °C). After that, we diluted 50 μL of the cell cultures in 1 mL of filtered cold Dulbecco's PBS (Sigma-Aldrich #D8537) in 15 mL FACS tubes.

We performed FACS-sorting on a FACS Aria III flow cytometer (BD Biosciences, San Jose, CA) with a 70 μm nozzle for droplet formation. We used a 488 nm laser to detect forward scatter (FSC) and side scatter (SSC) with a 488 nm/10 nm band-pass filter. We set the flow rate to 1.0 and diluted samples if necessary to obtain a cell count of approximately 5000 events/second. Because bacterial cells are very small, we decreased the limit of particle detection (threshold) to the minimum possible (200 arbitrary units on both FSC and SSC channels). If background noise (such as caused by particles from the medium) was too high, we increased this threshold to a maximum of 500 arbitrary units on both the FSC and SSC channels. Next, we measured FSC-H and SSC-H of cells containing our negative control plasmid and set our FSC and SSC voltages to have our bacterial population on scale (centered in the software visualization panel, Supplementary Fig. S4). We sorted and binned cells by sfGFP fluorescence (FITC channel, 488 nm laser, emission filters 502LP, 530 nm/30 nm). We chose the FITC channel voltage such that the median fluorescence of the negative control sample (promoterless pCAW-Sort-Seq plasmid, without sfGFP expression) was between 0 and 100 (arbitrary units) on the FITC-H axis.

We set sorting gates on the FITC-H axis as follows: First, we recorded autofluorescence of the negative control cell culture. The median autofluorescence from this control served as the upper boundary of the lowest bin (B1) for the experimental population. Then, we recorded the fluorescence of $10^6$ cells expressing sfGFP and harboring the library, without sorting. The choice of this number ($10^6$) of cells was based on the combination of library size and the estimated loss of library diversity post-sorting (a reduction of up to 70% in sequence diversity was found in similar sort-seq studies S17[28]). Thus, we estimated that 975,000 cells would be required to represent the whole library ($N = 65,536$), with at least 5 cells harboring a copy of each sequence and accounting for diversity loss. We then proceeded to set our binning gates. We took the lower bound of the highest bin (B13) for the experimental population to correspond to the 95th percentile of the fluorescence distribution of this population. We chose boundaries between intermediate bins with equidistant spacing on a binary logarithmic ($\log_2$) scale. After we had the gates in this way, we calculated the fractions of the previously $10^6$ recorded cells that were inside each gate.

We determined the number of cells to be sorted into each of the 13 bins from the fraction of cells we had previously recorded in each of the bins, such that the total number of sorted cells was equal to $10^6$. We sorted cells into 1.5 mL Eppendorf tubes with 500 μL of LB medium each. We kept the tubes cooled to 4 °C to halt growth while sorting, and during the sorting of subsequent samples. We carried out the sorting procedure in three replicates derived from three independent library transformations.

We added 1 mL of LB without antibiotics to each tube and removed 20 μL of the resulting volume for serial dilutions. We transferred the remaining liquid culture (980 μL) to 50 mL falcon tubes and allowed each culture to recover for 2 hours (37 °C, 220 rpm). After recovery, we added 9 mL of LB supplemented with chloramphenicol, and grew the cultures overnight for freezing them in glycerol stock aliquots, reassessing our binning procedure (see below), and extracting plasmid DNA for subsequent PCR and sequencing steps. We used the 20 μL initially removed from each 1 mL culture for preparing two serial dilutions ($10^{-4}$ and $10^{-6}$) that were plated on LB-Cm agar plates (200 μL per plate) to estimate the post-sorting viability through cfu counting. By knowing how many cells were sorted into each bin, we can estimate how many cells would be expected in our dilutions and compare this number with the CFU counts we observed. In this way, we estimated that on average (across bins) 73% of cells remained viable (standard deviation: 15%). We also estimated the genetic diversity of the library through Sanger sequencing of colonies (NightSeq® service, Microsynth, Switzerland).

For reassessing our binning procedure, we re-grew binned cultures (directly on the next day after sorting, using overnight recovered cultures, or by re-growing frozen aliquot stocks) and measured their expression distributions by flow cytometry. This procedure reproduced the original expression measurement (Supplementary Fig. S5). It also allowed us to ask whether the resulting fluorescence distributions had the same geometric mean as those we had previously obtained (before sorting) for each sorting gate. The geometric mean is often preferred in this type of analysis over the arithmetic mean, because it is less affected by the presence of outliers in the data[144,146]. It is also a more accurate representation of the central tendency of data that is log-normally distributed, which is often the case with flow cytometry fluorescence measurement[144,146].

**Library diversity.** We sorted cells harboring our library into 13 "bins" based on their fluorescence, and deep-sequenced TFBS variants from each bin (Methods, Supplementary Figs. S5 and S6). The resulting library comprised 48,937 genotypes, covering 75% of the $4^8 = 65,536$ genotypes within the genotype space we studied (Supplementary Fig. S7). Sequencing did not recover all $4^8$ genotypes, because of technical constraints common to mutational library studies, such as biases in library synthesis[147], PCR amplification[148], cloning, and loss of sequence diversity after cell sorting[28]. For all further analyses, we additionally applied a stringent sequencing depth threshold. That is, we analyzed only TFBS variants with at least 30 sequencing reads, which reduced our library size further to 17,851 genotypes (27% of $4^8$, Supplementary Figs. S7). Some previous studies have included genotypes with single read counts[28,39,40], but they focused on machine-learning-based "oracle" models for the prediction of expression levels[39,40]. We adopted a more conservative approach for greater reliability, because low read counts can compromise the accuracy of our analysis. For example, TFBS variants with one or few reads are subject to greater measurement noise and thus low accuracy of repression levels estimates. Including such variants might affect a study's conclusions.

### DNA extraction and DNA preparation for sequencing

We diluted 500 μL of individual glycerol stocks of each replicate sub-population of sorted cells (i.e., cells from each "bin" of fluorescent intensity) in 5 mL of LB supplemented with chloramphenicol in 15 mL Falcon tubes, and grew the resulting cell culture overnight (16 hours, 37 °C, 220 rpm). On the next day, we isolated plasmids from each culture using a QIAprep® spin miniprep kit (Qiagen, Germany). In order to allow the sequencing of multiple pooled samples (multiplexing), we barcoded our regulatory region through PCR with specific HPLC-purified primers (Supplementary Table S6) provided by Eurofins (Konstanz, Germany). We added barcodes only at the 5′region of the amplicon through a PCR reaction. We performed the PCR reaction with the Q5 high-fidelity polymerase in triplicate for each sample. We calculated the primer melting temperatures (Tm) following the NEB Tm

calculator (https://tmcalculator.neb.com/#!/main) for a primer concentration of 500 nM. We performed PCR with the following program: 98 °C/30 s; 25 cycles of 98 °C/10 s, 64 °C/30 s and 72 °C/30 s; and 1 cycle of 72 °C/2 min.

After PCR amplification, we digested the reaction products with both the restriction enzyme DpnI (NEB #R0176L) and Exonuclease I (NEB #M0293L) in order to remove traces of genomic DNA, plasmids, and single-stranded DNA that could interfere with sequencing. The Master Mix we used for a single digestion harbored 1 μL of 10× CutSmart® Buffer (NEB #B6004S), 1 μL of Exonuclease I (NEB #M0293L), 1 μL of DpnI restriction enzyme (NEB #R0176L), and 7 μL of distilled nuclease-free water. For each PCR product, we added 10 μL of the Master Mix. We incubated the reaction for 1 hour at 37 °C, following 15 minutes at 80 °C for deactivation of the enzymes.

After digestion, we purified samples using the Monarch® DNA PCR/Gel Extraction Kit (NEB #T1020L. We also analyzed samples through gel electrophoresis to confirm that a single band with a size around 150 bp was present for each of them. After confirming the expected bands, we pooled the samples for the different bins of each replicate equimolarly to a total mass of 2600 ng and a volume of 100 μL (26 ng/μL of DNA) in 1.5 mL Eppendorf tubes. We then sent the pooled samples for adapter ligation and sequencing at Eurofins (NGSelect Amplicons® on Illumina HiSeq), obtaining 15 million paired-end reads (2 × 150 bp) for all samples.

### Data analysis

**Filtering and preparing sequencing reads.** We processed the sequencing data using a combination of in-house Python and awk scripts, as well as standard bioinformatics tools. Briefly, we first trimmed sequences to remove Illumina adaptors, merged the paired reads, and separated them into different files according to their sequence barcodes, which reflect the specific fluorescence bins from which they originated, Supplementary Table S6.

In order to remove Illumina adaptor sequences from the paired-end reads (in both forward and reverse orientations), we used Cutadapt[149] with the following options:

*cutadapt -j 8 -e 0.1 --no-indels --overlap=8 --discard-untrimmed \\*
*-a "^\\$FWD...\\$REV_RC;max_error_rate=0.2;min_overlap=6" \\*
*-A "^\\$REV...\\$FWD_RC;max_error_rate=0.2;min_overlap=6" --pair-filter=any \\*
*-o 'Sample_${sample}_trimmed_1.fastq.gz' \\*
*-p 'Sample_${sample}_trimmed_2.fastq.gz' \\*
*--max-ee=2 -l114 \\*
*$reads*

The "read1" adapter we used was:
5'-AGATCGGAAGAGCACACGTCTGAACTCCAGTCA-3'
The "read2" Adapter we used was:
5'-AGATCGGAAGAGCGTCGTGTAGGGAAAGAGTGT-3'
Because our amplicons were short (153 bp) and the paired-reads thus fully overlap, we had to merge them. We performed this process with the software FLASH[150]. After trimming and merging the paired-end reads, we performed the demultiplexing process using the following FLASH options:

*flash -t $ncores $reads -O -m 60 -M 140 -z -o 'Sample_${sample}_merged'*

Subsequently, we used the open-source tool FastX (http://hannonlab.cshl.edu/fastx_toolkit/) to identify and retain only high-quality sequences (quality threshold of Q = 33).

After these steps we filtered our data further, retaining only sequences that showed neither mutations nor indels in the *sfgfp* regulatory region outside the variable TFBS library region. For this purpose, we developed an in-house python script. Subsequently, we used in-house *awk* and R[151] scripts to compile the data as a single table containing all unique sequences as rows, and the number of reads for

each bin as columns. We used this table to calculate the average gene expression level driven by each sequence.

**Calculating repression levels.** In sort-seq experiments, sequences can appear in more than a single fluorescence bin due to both random mis-sorting[64,81] and stochastic gene expression noise[142]. Therefore, following previous studies[39,40,48], we estimated expression levels for each TFBS sequence as the weighted average of bins in which the sequence was observed. Specifically, we multiplied, for each sequence, the number of times ($x_i$) that we had observed the sequence in each bin $i$ by an integer representing that bin ($w_i$=1, 2, 3, 4,...,13) and averaged these values by the total number of read counts for that sequence. In mathematical terms, we calculated the weighted average

$$e = \sum_{i=1}^{n} \frac{(x_i * w_i)}{\sum_{i=1}^{n} x_i}$$

to quantify the expression level e driven by the sequence.

This procedure resulted in a continuous distribution of expression levels e within the interval (1, 13). Since high expression values represent high GFP expression and, therefore, low-affinity binding of TetR to a TFBS variant, we reversed the scale to facilitate biological interpretation, where high values correspond to strong repression and low GFP expression. Specifically, we calculated

$$r = e_{max} + 1 - e$$

as a measure of the repression level r conferred by a sequence, where $e_{max}$=13 is the maximum possible value for the expression level. In addition, to obtain a metric that lends itself to more intuitive interpretation, we normalized the previously calculated average repression levels, dividing them by the wild type repression level. Hence, the repression scores we used in this study range from 0 to 1.25, where low values represent TetR TFBS variants that bind TetR with low-affinity, and high values represent variants that bind TetR with high-affinity and repress gene expression strongly. The value of one corresponds to the repression conferred by the "wild-type" *tetO2* binding site.

**Combining the data from triplicates.** Firstly, we removed from further analysis all sequence variants that were not present in all triplicates. Secondly, we eliminated all sequences that did not have a minimum of 30 reads in all 13 bins (Supplementary Fig. S8). We then calculated the repression levels (see *Calculating individual average expression levels and repression levels* section) for each remaining sequence in each replicate. Thirdly, we calculated the coefficient of variation of repression levels for each sequence (across replicates) as a measure of how widely distributed the repression levels were among replicates. We filtered out sequences with a coefficient of variation above 0.5. In doing so, we followed common practice in transcriptional studies using fluorescent reporters, which accept a coefficient of variation up to this magnitude as admissible due to transcriptional noise and technical variability of measurements[142]. Lastly, we combined our datasets by averaging across replicates the repression levels conveyed by each remaining sequence.

**Frequency matrix and sequence logo.** We generated frequency matrices of nucleotides for TetR TFBSs from each fluorescence bin by counting the frequency of each nucleotide at each variable position of the *tetO2* library. We generated heatmaps representing the frequency matrices and DNA sequence logos in R. A sequence logo consists of a stack of letters at each position of a DNA sequence, where the relative sizes of the letters indicate their frequency in the sequence. The total

height of the stack corresponds to the information content of that position, in bits[152,153].

Such logos are graphical representations of informational properties of DNA. When mutated, characters with high information content are more likely to lead to a loss of binding (repression) than characters with low information content.

**Creation of genotype networks and network metrics.** We used in-house Python script and the python package *igraph*[154] to generate a directed genotype network. This is a graph in which TetR TFBS variants are nodes (vertices, genotypes), and variants that differ in a single nucleotide are connected by an edge. We extracted the largest weakly connected subgraph (giant component[4]) of the network and used it for all further analyses. This giant component comprises the vast majority (97%) of sequenced genotypes. Each genotype is associated with its corresponding repression level. Each edge is directed, i.e., it corresponds to a repression-level-increasing mutation and points from the variant with a lower repression level to the neighbor with a higher repression level. We used in-house Python and R scripts for all network analyses described in the sections below.

**Epistasis.** Epistasis, non-additive interactions between mutations, can impose severe constraints on molecular evolution, because the mutations that are beneficial in one genetic background may be deleterious in another[75]. Epistasis can be classified as magnitude, simple sign, or reciprocal sign epistasis, depending on the sign (i.e., positive or negative) of the fitness effect of individual mutations and their combinations[75] —with increasingly detrimental effects on landscape navigability[108,109]. In magnitude epistasis, the effect of a mutation on repression varies depending on the genetic background but the sign of this effect (increasing or decreasing repression) does not. Simple sign epistasis occurs if one single mutant has a lower repression level than both the wild type and the double mutant, while the other single mutant has a repression level that is intermediate to the wild type and double mutant. Reciprocal sign epistasis occurs when both mutations independently decrease repression but their combination increases repression. Thus, the presence of reciprocal sign epistasis is a necessary condition for the existence of multiple peaks in an adaptive landscape[108,109].

To conduct epistasis calculations, we employed a method that involved identifying all "squares" in the genotype network using the *motifs* function from the *igraph* library in R. Each square consisted of a "wild-type" sequence, a double-nucleotide mutant, and the corresponding two single mutants (see Supplementary Fig. S16 for more information on epistasis squares). We assessed epistasis for each square along a single axis by selecting the highest-repression sequence as the double mutant. Our analysis categorized the landscape into three groups: no sign epistasis, simple sign epistasis, and reciprocal sign epistasis, which we explained in detail in Supplementary Table S1 and Supplementary Fig. S16. The no sign epistasis category included both magnitude epistasis and additivity (no epistasis), without differentiating between them, as neither affected peak accessibility[4,155]. Finally, we determined the proportion of all squares whose constituent mutations interacted based on these classifications (see Supplementary Table S1 and Supplementary Fig. S16).

**Peaks and plateaus.** A peak is a genotype (TetR binding site variant) whose neighbors all convey lower repression than itself. Two peaks are connected if they are neighbors and convey the same repression. Connected peaks are individually characterized as peaks and together as a "plateau". We refer to the genotype with the highest repression level as the summit or global peak[4,155].

**Accessible paths.** A mutational path through the genotype network is accessible, if and only if the repression level increases for each

mutational step along the path[4,155]. We enumerated accessible paths of all lengths (mutational steps) exhaustively.

**Basins of attraction.** The basin of attraction of a peak comprises all TetR binding site variants from which accessible paths to the peak exist. We refer to the basin's size as the number of variants in the basin. We determined basin sizes by exhaustive enumeration.

**Overlap between basins.** The basins of attraction of different peaks may comprise overlapping sets of variants. To determine the overlap between two basins $B_1$ and $B_2$, we used the Jaccard index $J$[113,156], which is equal to the size of the intersection between two sets of variants divided by the size of their union:

$$J = \frac{B1 \cap B2}{B1 \cup B2}$$

**Generating randomly shuffled landscapes.** To generate an uncorrelated random landscape based on our TetR landscape, we randomly permuted repression strengths among our genotypes (TFBSs). This random shuffling randomizes the assignment of regulation strengths to genotypes while preserving their original distribution. We then rebuilt the landscape from the shuffled data, and identified its peaks, using the same methods as for the original data (Methods). To ensure statistical robustness, we repeated the random shuffling and network analysis 1000 times. We then compared the average number of peaks from the 1000 shuffled landscapes to that from the original landscape. Shuffling resulted in an average of 2721 ± 92 peaks (mean ± s.d.), with an upper bound of 2872 peaks at a 95% confidence level. The upper bound is defined as the value below which 95% of the observations fall.

**Principal component analysis.** In order to investigate the proximity of peaks and other variants in sequence space, we prepared our data with a one-hot encoding method. A one-hot encoding represents each categorical value (nucleotide at each position of a DNA string) as a binary vector of length 4. It thus converts an entire DNA string of length L into a 4xL binary matrix. We used the binary matrix to perform the PCA analysis. We performed the PCA analysis using the R base function *prcomp()*.

The sequence space underlying our landscape is high-dimensional. Like other spaces in which DNA sequences can harbor more than two alleles (A,T,G,C) per site, it forms a Hamming graph[157]. In contrast, spaces with two alleles per site form a simpler topology called a hypercube, a special case of a Hamming graph[157]. Although dimensionality reduction techniques like PCA are applied in the visualization of genotype spaces[113,158,159], their effectiveness in representing combinatorially complex spaces, such as Hamming graphs, can be limited due to minimal redundancy in genetic information across different dimensions[160]. In consequence, each principal component explains only a small amount of genetic variation. For example, PC1 explains only 5% of the variance in our data. Another consequence is that the clustering of peaks may depend on whether they share some allelic states. To illustrate the differential clustering of peaks, we show projections on the first five components of the PCA (Supplementary Fig. S16).

## Validating repression levels with plate reader measurements

To further validate our calculated repression levels, we plated cells from each fluorescence bin onto LB-agar plates, picked 15 colonies of cells from each bin (N = 195), sequenced their TFBS sequence, and replated them onto LB-agar plates. We picked individual colonies and grew them overnight (16 hours, 37 °C, 220 rpm) in liquid LB supplemented with 50 µg/mL of chloramphenicol. We diluted the cultures to 1:10 (v/v) in cold Dulbecco's PBS (Sigma-Aldrich #D8537) to a final

volume of 1 mL. We transferred 200 μl of the diluted cultures into individual wells in 96-well plates and measured GFP fluorescence (emission: 485 nm/excitation: 510 nm, bandpass: 20 nm, gain: 50) as well as $OD_{600}$. We then normalized fluorescence by $OD_{600}$ measurements, and compared the obtained ratios to the previously calculated repression levels for the 195 selected variants. We observed a strong negative Pearson correlation ($R = -0.86$, Fig. 2c). We performed all such measurements in biological and technical triplicates (three colonies per sample and three wells per colony, respectively).

## Simulated adaptive walks

We simulated the adaptive evolution of a population on the adaptive landscape by performing three different types of random walks, a "greedy adaptive" random walk, a "uniform adaptive" random walk, and a random walk based on Kimura's model of fixation probability. Our simulations also account for mutation bias, which means that different types of mutation have a different probability of occurring in a population[161]. We use mutation biases that were experimentally determined for *Escherichia coli*[162,163].

For all simulations, we assumed that only point mutations occur and that the time it takes for a point mutation to become fixed in a population is much shorter than the time it takes for a new mutation to appear that will eventually go to fixation. This scenario is also referred to as the SSWM scenario[40,90–92]. Under this scenario, evolving populations are monomorphic most of the time, i.e., all individuals have the same genotype. This scenario is realistic when the product of effective population size $N$ and mutation rate $\mu$ is small ($N\mu < 1$), which is the case for *E. coli* ($N = 1.8 \times 10^8$, $\mu = 2 \times 10^{-10}$)[89]. In this scenario, adaptive evolution on a fitness landscape can be modelled as an adaptive random walk.

In the greedy adaptive random walk, we assume that only the mutation that conveys the largest fitness advantage becomes fixed. We initiated one greedy random walk from each non-peak genotype and terminated the walk once no genotype with higher fitness was reachable, i.e., once it had reached a fitness peak. A greedy random walk is deterministic. It is thus sufficient to initiate one greedy random walk per starting genotype.

In the uniform adaptive random walk, we relax the conditions of the greedy random walk by allowing all fitness-increasing mutations to fix with equal probability. We randomly chose 1000 non-peak starting genotypes and simulated 1000 uniform random walks for each of these starting genotypes. Random walks terminate when they reach a fitness peak.

Both types of adaptive walks ignore the possibility of genetic drift, which allows mutations with no or negative fitness effects to become fixed in a population. A well-established model for random walks that permit genetic drift uses fixation probabilities computed by Kimura[88,95,96] i.e., $f_{ij} = (1 - e^{-2s})/(1 - e^{-2Ns})$, where $f_{ij}$ is the probability of fixing mutation $j$ in the background of genotype $i$, $N$ is the effective population size, and $s$ is the selection coefficient, i.e. the difference in repression strengths between genotypes $i$ and $j$[88,96]. For a given pair of genotypes, the only parameter of this model is the effective population size $N$, for which we explored values of $N = 10^8$, $N = 10^5$ and $N = 10^2$. Generally, the smaller the population size, the larger the probability that neutral or deleterious mutations become fixed in a population.

For these "Kimura" adaptive walks, we chose 1000 random starting genotypes for each of the three values of $N$, and simulated 1000 random walks for each of the starting genotypes. At each step, we randomly picked a mutation $j$, generated a random number on the interval [0, 1], and considered the mutation to become fixed if the random number fell within the interval [0, $f_{ij}$]. Computationally, we accelerated this process by precomputing all fixation probabilities for all genetic backgrounds, and then used the *Python* function *numpy.choice* to generate a random sample from the multinomial distribution of fixation probabilities at each step[164]. Because of genetic drift,

Kimura's random walks do not necessarily terminate when they reach a fitness peak. For this reason, we simulated each such random walk for 1000 mutational steps, which allows multiple fitness peaks to be visited.

## Noise sensitivity analysis

Fluorescence-based sorting methods are inherently noisy, especially at the highest and lowest bounds of the fluorescence distribution[64,81,82]. To account for the experimental noise in our data and to study its effects on landscape topography and adaptive evolution, we computed an individual noise value (tau, τ) for each genotype that is equal to the standard deviation of its repression strengths across the three replicates. We then recreated the genotype network underlying our adaptive landscape, assigning the directionality of edges connecting each pair of neighboring genotypes A and B based on this noise.

Specifically, for every genotype A and each of its single mutant neighbors B, we considered that the repression strength $S_B$ of B is different from the repression strength $S_A$ of A if the following conditions are met for the noise values $\tau_A$ and $\tau_B$ of the two genotypes.

First, we consider the repression strength of A higher than that of B if

$$S_A > S_B + \tau_B$$

and if

$$S_A - \tau_A > S_B + \tau_B$$

In this case, we create a directed edge from A to B.

Second, we consider the repression strength of A smaller than that of B if

$$S_B > S_A + \tau_A$$

and if

$$S_B - \tau_B > S_A + \tau_A$$

In this case, we create a directed edge from B to A.

If neither of these pairs of conditions are met, we consider the repression strengths of A and B indistinguishable. In this case, we create two directed edges, one from A to B, and another from B to A.

In addition, we performed a sensitivity analysis by broadly varying the amount of noise in the data. In this sensitivity analysis, we increased the experimentally observed noise value τ for each TFBS repression strength by 10%, 25%, or 50%, yielding simulated noise values that we call $\tau_{1.1}$, $\tau_{1.2}$, and $\tau_{1.5}$, respectively, Supplementary Fig. S21.

We first used this noise to explore how varying noise affect the landscape's features, particularly the number of peaks, their organization into plateaus, and the accessibility of high peaks. For counting peaks and plateaus, we adopted the definitions in Supplementary Material 9.5. We observed that increasing noise (i) reduces the number of peaks (Supplementary Fig. S21a) because the added noise smooths out minor fluctuations in the landscape, leading to fewer distinct local maxima; (ii) increases the accessibility of high peaks (Supplementary Fig. S21b) because noise can create additional paths to high peaks or lower barriers between them, making it easier to reach high fitness peaks; (iii) increases the number of plateaus (clusters of connected neighboring peaks, Methods, Supplementary Fig. S21c) because noise causes neighboring peaks to merge into broader plateaus of indistinguishable fitness, reducing the separation between peaks; and (iv) increases mean peak "breadth" (Supplementary Fig. S21d), because the merging of peaks into plateaus results in each plateau encompassing a

larger number of peaks. We define peak breadth as the number of peaks composing a plateau. Mean peak breadth is the average number of peaks in a plateau.

Given that the sensitivity analysis revealed substantial changes in landscape properties, we also asked how these changes would impact adaptive evolution. To this end, we conducted Kimura random walks on landscapes with varying experimental noise levels (Methods) that range from no noise to 150% of the actually observed noise. We performed these simulations at each noise level for populations of $10^8$ individuals and for the same 1000 randomly selected initial genotypes (Methods, Supplementary Fig. S21). We found that the percentage of adaptive walks (out of $10^6$) reaching high peaks increases as noise increases (Supplementary Fig. S22). It increases to 37% at the highest noise level. This increase is a consequence of landscape smoothing caused by noise (Supplementary Fig. S21). Notably, this analysis also shows that noise does not strongly affect the number of times a genotype is visited across all adaptive walks. That is, this number is highly correlated between landscapes with and without noise (Supplementary Fig. S22). Similar genotype visitation frequencies indicate that evolutionary dynamics at the genotype level is not strongly affected by noise.

### Reporting summary

Further information on research design is available in the Nature Portfolio Reporting Summary linked to this article.

## Data availability

The DNA sequencing data generated in this study have been deposited in the NCBI database (BioProject) under the accession code: PRJNA1019339. The flow cytometry and plate reader data generated in this study have been deposited in the Zenodo public repository and are accessible publicly via the following https://doi.org/10.5281/zenodo.8370874. Source data are provided with this paper.

## Code availability

The computer code generated in this study has been deposited in the Zenodo public repository and is publicly accessible via the following https://doi.org/10.5281/zenodo.8370874.

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

## Acknowledgements

We gratefully acknowledge financial support from the Swiss National Science Foundation grant 310030_208174. We also extend our thanks to the UZH University Priority Research Program in Evolutionary Biology and the UZH flow cytometry facility for their technical support. We are especially thankful to Diego Pesce for his assistance in establishing experimental protocols, and to Andrei Papkou for his guidance with computational analysis and for engaging in theoretical discussions.

## Author contributions

C.A.W. and A.W conceived the study and designed experiments. C.A.W. carried out experiments. C.A.W. wrote computer code to carry out bioinformatic work and data analysis. A.W. provided analytical guidance in data analysis. C.A.W. generated figures. L.G. wrote computer code to carry out simulations. C.A.W. and A.W. wrote the paper, which was edited by all authors.

## Competing interests

The authors declare no competing interests.
