## [Peer Review file · Nature Communications]

The highly rugged yet navigable regulatory landscape of the bacterial transcription factor TetR

Corresponding Author: Professor Andreas Wagner

Version 0:

Reviewer comments:

Reviewer #1

(Remarks to the Author)

The manuscript reports a fluorescence-based assay of the effects on repression of several thousand mutants of the *Escherichia coli* tetO2 transcription factor binding site. After the generation of tetO2 libraries by using oligonucleotides with 8 rationally selected degenerate positions, cells were sorted using FACS into 13 bins according to fluorescence intensity, which inversely correlates with repression strength. The sequences at the degenerate positions were sequenced and analysed by bin, and repression strength could then be associated with the sequence. The authors were then able to draw a network representing the fitness landscape of these mutants, where fitness was equal to repression strength. The resulting fitness landscape of repression is highly rugged and moderately navigable, with high repression peaks having generally larger basins than lower peaks. However, since there are many more lower peaks, fewer genotypes can attain high peaks. This was confirmed using simulation based on several regimes, which all converged to roughly the same results, with most often ~20% of populations reaching high peaks. Which specific peak was reached appears to be highly stochastic. This represents the largest genotype/phenotype association analysis for a prokaryotic transcription factor binding site to date, as well as a thorough computational work simulating the evolution of the genotype/phenotype towards high repression. This work is relevant in the methods that it presents to study the effects of mutations on prokaryotic transcription factor binding sites, as well as the evolution of the TetR regulation landscape evolution towards high repression.

The experimental and computational methodologies are sound, as well as the analyses and general conclusions from the results. Several conclusions were interesting and unexpected, such as the broad distribution of high fitness peaks in genotype space and, the relationship between the shortest accessible path between peaks and the mean genetic distance between variants and peaks. The prevalence of reciprocal sign epistasis was also surprising. The use of several regimes in modelling, including neutrality, strengthens the conclusions drawn in this part of the paper.

However, while the analyses are in-depth, some are narrow, focusing on a few aspects of the system, landscape, and model, and would benefit from more in-depth analyses of other aspects and biological insights that can be gained from them. For the moment, much effort is spent on characterizing the fitness landscape without trying to connect it to the actual biology of the system (fitness effects, probability of mutation types).

Assumptions are made about the "fitness" of the TetR system that take it out of its biological context and relevance. Higher repression does not equate to higher fitness in all conditions. It seems that this is something that would have been possible to measure using pooled competition assays. Studying evolvability and accessibility in the landscape of a system in which total, non-modulable repression can mean cell death, such as the presence of tetracycline, without considering the plasticity of the landscape, narrows the biological relevance of the conclusions.

Here are general comments on what would have strengthened the manuscript:

1) State clearly and early on that in this study, fitness is considered as being the strength of repression, or replace fitness by strength of repression. Using fitness to describe what is studied in this paper disregards the biological role and context of the tet system, especially since the actual effect on fitness from these mutations is not measured, neither in the absence nor presence of tetracycline. Since the WT repression is strong, one can imagine that fitness somewhat correlates with stronger repression. Beyond this, the relationship between expression and fitness is speculative. It is mentioned that there are reports that overexpression of tetA in the absence of tetracycline is deleterious, it is not explored or explicated enough to support the use of fitness to describe the metric measured in this study. While it does not invalidate the conclusions, this assumption about fitness in this system taints the rest of the analyses and conclusions. In the discussion, in line 545-546, it is said that "More generally, the assumption that the strongest regulation is the most beneficial may not always apply." This should absolutely be part of the introduction to give perspective on the biological relevance of the system used. The concepts

introduced in paragraphs from lines 548 to 559 should also be introduced earlier in the paper, as these are key concepts about the biological relevance of the results.

2) Since the presence of antibiotics in the environment is going to be variable, it seems that the repression system would be optimized according to the frequency of its presence. For instance, one can imagine that a permanent low concentration would favor a different level of repression than a very rare strong concentration. It seems that the measurement of the link between fitness and repression with and without antibiotics would have helped make the evolutionary simulations more by guiding the choice of conditions to simulate evolution in.

3) A big point of contention is the absence of over 70% of the theoretical sequences generated in the experiment. 65 536 mutants were generated, but only 17 851 were detected in the experiment. The fact that replicates correlate strongly only increases confusion since these almost 40,000 mutants appear to be absent in all replicates. No explanation is given for this. Lack of initial coverage? Issues in oligo library synthesis? Is it from a limitation in the method, or does it have biological relevance, like a dominant negative effect? This issue should be discussed at some point in the main text. It is mentioned in M&M that this is likely associated with PCR amplification, but this could probably have been avoided by using several technical replicates and post-reaction mixing.

4) A more in-depth analysis of the resulting network could have been done. Investigating the connectivity of detected nodes could have given insights into whether some regions of the network and, therefore, some closely related combinations of genotypes are missing. This could have been informative on why so many mutations are missing. On the other hand, if the standard deviation of connectivity is low, this means that the space was properly sampled and that the missing mutants come from methodological limitations.

5) The notation of +/- s.d. should be standardized. See, for instance, lines 399-400 and 401.

6) The fact that some promoter sequences are stronger repressors than the WT is an interesting observation, but these results are not validated further, and these could stand out just because of noise in the measurements. Are differences statistically significant? Cell sorting is probably noisier in the lower expression bins, so it would be important to validate this result using another approach. It seems that from Figure 2c, the authors are set up to perform validation experiments.

7) The measure of reproducibility is presented as a correlation of read counts of each variant (Figure S7). While this is useful, it just represents technical reproducibility in the library construction process. What is critical is the correlation between the three replicates in the estimated repression levels of individual variants, i.e. in the measures that are used to make inferences on the adaptive network. It would also be important to know how measurement errors impact the overall structure of the network, the identity of the peaks and, for instance see point 6 above. Experimental noise is likely not equally distributed among the expression bins, so is it possible that measurement noise itself impacts the navigability estimates of the landscape?

8) It was difficult to understand the role of the PCA analysis in the paper and what it represents, especially given that the top two PCs explain only 5% of the variation. Also, the sequence logo in Figure 2d seems to confirm this, with one preferred nucleotide at each position and most alternatives generally not tolerated.

9) I could not see in the simulation data the different types of mutations (transition and transversions). As far as I understand, for now, mutational neighbours are just one nucleotide change away. However, if mutations do not have the same rates, this could change conclusions regarding the navigability of this network.

Minor comments:

Line 62: "The molecular origins and evolution of such elements are still poorly understood." This sounds like a very strong statement.

Reviewer #2

(Remarks to the Author)

The authors present a large genotype-phenotype map that relates an 8-letter subsequence of a bacterial transcription factor binding site to the repression level induced by the tetracycline repressor TetR. The resulting landscape is analyzed with respect to its ruggedness (number and distribution of peaks) and navigability (existence of monotonic paths to peaks, basins of attraction of peaks, reachability of peaks by various types of evolutionary dynamics). The methodology appears to be sound, and the methods and results are clearly described. There are few (if any) data sets of comparable size and quality in the current literature, and the work therefore constitutes an important advance in the field that is principally well suited for publication in Nature Communications. However, I have some concerns about the analysis and interpretation of the results that the authors should address before I can recommend publication.

MAJOR CONCERNS

1. The studied landscape comprises only about 27% of the 65536 possible sequences. Since the TetR repression level does not affect the viability of the bacteria, my understanding is that this should be considered as a random, unbiased sample of the full landscape, the sparsity of which is mainly determined by the chosen sequencing depth (Fig. S8). However, the authors do not present any evidence that the sample is indeed unbiased. If it is, the resulting genotype network should be a random subgraph of the full Hamming space. The statistical properties of such subgraphs are well known in computational biology (see, e.g. work by Stadler and Reidys). For example, the degree distribution is expected to be binomial. The authors should analyze the structure of their genotype network along these lines, and if possible also study the effects of varying the sequencing depth threshold (and thus the sparsity of the network).

2. The fact that the landscape is a (sparse) random sample has important implications for the interpretation of the results, in particular for the identification and analysis of peaks. There are two opposing effects. On the one hand, a large fraction

(73%) of peaks are not sampled. On the other hand, genotypes are falsely identified as peaks, because they have neighbors of higher repression level that are not sampled. It is possible (and can be shown for uncorrelated random landscapes) that these effects partially cancel, such that the number of (mis-)identified peaks is similar to the number of peaks of the full landscape; indeed, the expected number of peaks for the full landscape with randomly assigned repression levels is 2621 according to a formula first derived by Kauffman and Levin (JTB 1987), not much larger than the number reported here (2092).

To put their estimate for the number of peaks into perspective, the authors could simply place uncorrelated random numbers onto the measured genotype network and determine the number of peaks of this random landscape. This should provide an upper bound on the ruggedness (compare to Ref.36). I expect that this bound will not be much larger than their actual estimate, supporting and quantifying their claim that the measured landscape is highly rugged. It would also be of interest to classify the genotypes identified as peaks by the number of neighboring genotypes that belong to the network, in order to determine how many are true peaks of the full landscape, and whether these tend to have high expression levels.

3. The authors argue that the fact that high repression peaks are more accessible than low peaks is "striking" and exemplifies a property "typically associated with smooth landscapes" (lines 524-527). I disagree with this statement, and actually believe it would be hard to come up with a fitness landscape where peak accessibility (measured in terms of quantities like the size of the basin of attraction or the number of accessible paths) is not correlated with peak height. In particular, such a correlation is well documented and understood for the null model with uncorrelated random fitness landscapes [often referred to as the House-of-Cards (HoC) model], which would not typically be considered a model with high navigability. Recent work on HoC landscapes has shown that the probability of existence of an accessible path shifts sharply from 0 to 1 at a (computable) threshold value of the fitness difference between the initial and final genotype [see Schmiegel and Krug, J. Math. Biol. 2023, as well as arXiv:1903.11913]. The threshold depends on the distance between the genotypes (scaled with the length of the genetic sequence) but remains nontrivial even at maximal distance. This implies that the fraction of genotypes from which a given peak genotype can be accessed increases continuously with the height of the peak, once the latter is large enough. The correlation between peak height and accessibility is therefore highly generic, and as such does not necessitate a "refinement of current landscape theory" (line 580/581).

OTHER ISSUES

4. line 166: "absence of the inducer". It is not mentioned in the main text that the "inducer" is Atc. This should be explained here or in the caption of Figure 1.

5. Figure 3d, lines 265-268: One of the two clouds appears to contain more peaks than the other. Which one is it?

6. Figure 4c: The grey shade around the black curve is hard to see.

7. Figure 4f: I don't quite understand this panel. How are the genotypes ordered in this plot? What is the origin of the blue cross? Also, in the caption it is stated that a value of zero is represented in yellow color, whereas in fact it appears to be blue.

8. line 380, Figure 5: What is meant by "all possible paths (both accessible and inaccessible)"? The total number of all paths between two genotypes (direct and indirect) grows double exponentially with genetic distance, and even the number of direct paths grows faster than exponentially (for two genotypes at distance d , it is $d!$). For $d=8$ there are 40320 direct paths, which is off the scale of Fig. 5b. Please clarify.

9. lines 421-428: How are the selection coefficients computed when evaluating the Kimura fixation probability? Is the normalized repression level simply interpreted as fitness? If so, this should be said.

10. line 465: assessed -> accessed? In the same line, the reference to Figure 5b appears to be incorrect.

11. Figure 5d, Figs. S15, S16: All these figures show a discontinuity in the cumulative frequency of adaptive walks around the wild type repression level. Where does this come from?

12. Some references to the bibliography appear to be incorrect. For example, on line 237, Ref.51 (Blount et al.) does not seem to be the correct reference for the giant component of a network, and on line 607 of the Supplement a reference 108 is cited that does not exist.

Version 1:

Reviewer comments:

Reviewer #1

(Remarks to the Author)

The authors have addressed all of my comments. Congrats on this nice piece of work!

(Remarks on code availability)

Reviewer #2

(Remarks to the Author)

The issues raised in my previous report have been fully addressed, and I recommend publication of the manuscript in its present form. There are three very minor points that should be fixed prior to publication:

1. In the caption of Figure 5b, I'm a bit puzzled by the information about the total number of paths ("The total number of paths is $N = 47,950,663$ "). Do I understand correctly that this is the number of paths of any length, originating at some variant and reaching some peak? If so, this should be specified more clearly. But in fact I don't really see the importance of this information; it may as well be omitted.
2. There is a typo in the caption of Figure 3 (nom-peak -> non-peak).
3. Ref. 85 has meanwhile been published (J. Stat. Mech. (2024) 034003).

(Remarks on code availability)

Dear Reviewers

Thank you for your largely positive evaluation of this manuscript, and for your thoughtful and constructive comments. We are grateful for the opportunity to improve this manuscript.

In response to the reviewers' comments, we performed new experiments and multiple new analyses. We also edited the manuscript extensively. The new observations we incorporated leave our key observations from the first submission unaffected. Below is our point-by-point response to the reviewer's comments. The comments are shown in black and our response in blue.

REVIEWER COMMENTS

Reviewer #1 (Remarks to the Author):

The manuscript reports a fluorescence-based assay of the effects on repression of several thousand mutants of the Escherichia coli tetO2 transcription factor binding site. After the generation of tetO2 libraries by using oligonucleotides with 8 rationally selected degenerate positions, cells were sorted using FACS into 13 bins according to fluorescence intensity, which inversely correlates with repression strength. The sequences at the degenerate positions were sequenced and analysed by bin, and repression strength could then be associated with the sequence. The authors were then able to draw a network representing the fitness landscape of these mutants, where fitness was equal to repression strength. The resulting fitness landscape of repression is highly rugged and moderately navigable, with high repression peaks having generally larger basins than lower peaks. However, since there are many more lower peaks, fewer genotypes can attain high peaks. This was confirmed using simulation based on several regimes, which all converged to roughly the same results, with most often ~20% of populations reaching high peaks. Which specific peak was reached appears to be highly stochastic. This represents the largest genotype/phenotype association analysis for a prokaryotic transcription factor binding site to date, as well as a thorough computational work simulating the evolution of the genotype/phenotype towards high repression. This work is relevant in the methods that it presents to study the effects of mutations on prokaryotic transcription factor binding sites, as well as the evolution of the TetR regulation landscape evolution towards high repression.

The experimental and computational methodologies are sound, as well as the analyses and general conclusions from the results. Several conclusions were interesting and unexpected, such as the broad distribution of high fitness peaks in genotype space and, the relationship between the shortest accessible path between peaks and the mean genetic distance between variants and peaks. The prevalence of reciprocal sign epistasis was also surprising. The use of several regimes in modelling, including neutrality, strengthens the conclusions drawn in this part of the paper.

However, while the analyses are in-depth, some are narrow, focusing on a few aspects of the system, landscape, and model, and would benefit from more in-depth analyses of other aspects and biological insights that can be gained from them. For the moment, much effort is spent on characterizing the fitness landscape without trying to connect it to the actual biology of the system (fitness effects, probability of mutation types). Assumptions are made about the "fitness" of the TetR system that take it out of its biological context and relevance. Higher repression does not equate to higher fitness in all conditions. It seems that this is something

that would have been possible to measure using pooled competition assays. Studying evolvability and accessibility in the landscape of a system in which total, non-modulable repression can mean cell death, such as the presence of tetracycline, without considering the plasticity of the landscape, narrows the biological relevance of the conclusions.

Here are general comments on what would have strengthened the manuscript:

1) State clearly and early on that in this study, fitness is considered as being the strength of repression, or replace fitness by strength of repression. Using fitness to describe what is studied in this paper disregards the biological role and context of the tet system, especially since the actual effect on fitness from these mutations is not measured, neither in the absence nor presence of tetracycline. Since the WT repression is strong, one can imagine that fitness somewhat correlates with stronger repression. Beyond this, the relationship between expression and fitness is speculative. It is mentioned that there are reports that overexpression of tetA in the absence of tetracycline is deleterious, it is not explored or explicated enough to support the use of fitness to describe the metric measured in this study. While it does not invalidate the conclusions, this assumption about fitness in this system taints the rest of the analyses and conclusions. In the discussion, in line 545-546, it is said that “More generally, the assumption that the strongest regulation is the most beneficial may not always apply.” This should absolutely be part of the introduction to give perspective on the biological relevance of the system used. The concepts introduced in paragraphs from lines 548 to 559 should also be introduced earlier in the paper, as these are key concepts about the biological relevance of the results.

We clearly are at fault here for not sufficiently emphasizing that we did not actually measure fitness. Thank you for pointing that out. We have corrected this oversight. We fully concur with your observation that, despite a common assumption in TFBS evolution modelling¹⁻⁴, the association between transcription factor binding site strength and fitness is not always linear and tends to be context-dependent. This complexity is well documented in the literature⁵⁻¹⁰, which highlights the non-linear and system-specific relationships between genotypes, phenotypes, and fitness. In relation to TetR and similar local bacterial repressors, we recognize that strong repression is typically favored to mitigate the costs associated with gene expression leakage¹¹, such as that from the TetA efflux pump.

It's important to note that our primary focus was on the phenotypic outcomes (specifically, GFP expression levels) rather than on fitness per se. Measuring fitness directly for tens of thousand TFBS variants is an important goal for future work. However, it would have required an entirely different experimental design that is beyond the scope of this revision. And it would not have provided information on the strength of gene regulation by individual operator variants. We added this clarification on our **introduction, lines 84-89**, as follows:

Whenever strong repression is associated with high fitness, the resulting regulatory landscape becomes a fitness landscape. We model the evolutionary dynamics of populations subject to Darwinian evolution on such a landscape. However, we emphasize that we did not measure fitness. Instead, we measured repression strength and modeled adaptive evolution under the assumption that repression strength can be a proxy for fitness.

We also gladly followed your suggestion by revising the manuscript to emphasize that we study repression strength rather than fitness. These revisions include cautionary notes added to both

the **introduction (lines 91–107)** and the **discussion (lines 654–670)**. For example, on lines **lines 91–107** we now write:

Existing evidence suggests that strong repression can indeed be associated with high bacterial fitness in the TetR system^{12–14}. For example, in the natural TetR system, strong repression of the efflux pump gene tetA by TetR is crucial for maintaining high fitness in the absence of antibiotics. Even low levels of TetA expression can impose a substantial fitness cost under these conditions^{12–14}, and most mutations in the tetO2 binding site reduce its affinity for TetR, and thus also fitness.

However, environmental changes often modulate mutational effects in gene regulation^{13,15–17}. For example, strong repression by TetR is not beneficial in environments with constant presence of the antibiotic tetracycline or with continuous shifts between the presence and absence of this antibiotic¹³. Moreover, the relationship between genotypes and fitness is not necessarily monotonic⁸. That is, the strongest regulation is not always most beneficial^{8,18,19}.

The relationship between repression strength and fitness is most likely to be strong and monotonic in environments where the antibiotic may be present but only rarely so¹³. Our evolutionary analysis of the TetR regulatory landscape should be interpreted with these caveats in mind.

And on **lines 667-670** we now write.

An advantage of our approach to mapping a gene repression landscape, rather than a fitness landscape, is that it allows the study of evolution under various relationships between repression strength and fitness. For instance, fitness might be highest at intermediate or low repression strengths.

2) Since the presence of antibiotics in the environment is going to be variable, it seems that the repression system would be optimized according to the frequency of its presence. For instance, one can imagine that a permanent low concentration would favor a different level of repression than a very rare strong concentration. It seems that the measurement of the link between fitness and repression with and without antibiotics would have helped make the evolutionary simulations more by guiding the choice of conditions to simulate evolution in.

Thank you for this insightful observation. The point is well taken. Although previous studies on the TetR operon have demonstrated substantial fitness reductions due to leaky expression of TetA^{14,20,21}, this fitness effect can indeed change depending on the environmental conditions. A previous experimental study with the TetR system revealed strong epistatic interactions influenced by environmental conditions, underscoring the inherent disadvantages of using an inducible system in varying regimes of antibiotic presence and absence¹³. In agreement with your comment, the same study has also shown that inducible genotypes undergo a prolonged lag phase when transitioning from an antibiotic-free environment to one containing antibiotics¹³. Additionally, in environments with sustained antibiotic presence, the complete induction of resistance in inducible genotypes is inhibited due to the continuous activity of their repressor.¹³ In sum, there are several additional reasons that support your point. We now elaborate in them in the revised **discussion** as follows (**lines 654-670**):

The evolutionary analysis of the TetR regulatory landscape is complicated by the fact that the relationship between repression strength and fitness can depend on the environment^{15,16}. A

previous experimental study demonstrated strong epistatic interactions between the mode of gene regulation of the tetracycline resistance tet operon (inducible by TetR or constitutively expressed) and the promoter strength of the resistance gene (low, medium, or high)¹³. These interactions are influenced by different tetracycline concentrations, suggesting that determining the optimal mode of expression (constitutive or inducible) requires knowledge of the promoter strength of the tet operon¹³. The same study also showed that inducible antibiotic gene expression may be detrimental in environments with fluctuating antibiotic presence. Notably, populations in which the tet operon is tetracycline-inducible experience a prolonged lag phase when transitioning from an antibiotic-free environment to one containing antibiotics¹³. Additionally, in environments with sustained antibiotic presence, the complete induction of resistance renders inducible expression less efficient than constitutive expression, because TetR inhibits full expression of the resistance operon¹³. An advantage of our approach to mapping a gene repression landscape, rather than a fitness landscape, is that it allows the study of evolution under various relationships between repression strength and fitness. For instance, fitness might be highest at intermediate or low repression strengths.

Furthermore, we have expanded our discussion and now point out that the adaptive landscape constructed in our study could be adapted to explore selection for low or intermediate regulatory strengths (**discussion, lines 667-670**).

An advantage of our approach to mapping a gene repression landscape, rather than a fitness landscape, is that it allows the study of evolution under various relationships between repression strength and fitness. For instance, fitness might be highest at intermediate or low repression strengths.

3) A big point of contention is the absence of over 70% of the theoretical sequences generated in the experiment. 65 536 mutants were generated, but only 17 851 were detected in the experiment. The fact that replicates correlate strongly only increases confusion since these almost 40,000 mutants appear to be absent in all replicates. No explanation is given for this. Lack of initial coverage? Issues in oligo library synthesis? Is it from a limitation in the method, or does it have biological relevance, like a dominant negative effect? This issue should be discussed at some point in the main text. It is mentioned in M&M that this is likely associated with PCR amplification, but this could probably have been avoided by using several technical replicates and post-reaction mixing.

Thank you for pointing out that we did not clarify the reasons for the incomplete sampling of genotypes in our landscape. We now explain these reasons in much more detail. A first reason is that we performed three independent transformations with the same pool of library DNA to obtain our biological replicates. This approach may have introduced biases from earlier steps (e.g., synthesis, PCR amplification, and cloning) and contributed to the incomplete sampling.

Secondly, for all analysis (including the correlation analysis between the read counts across replicates), we only considered sequences present in all three replicates. This decision was made to ensure consistency and reliability in our data analysis. Thirdly, while technical biases in library synthesis, amplification, cloning, and sorting contributed to reduced diversity, the most significant factor in diversity loss was our stringent quality control measures. We focused on sequences with a minimum of 30 paired-end reads in each replicate to ensure reliable estimates of regulatory strength. We chose this threshold to avoid the risk of low read counts leading to misassignment of repression levels, which could have biased our analyses. If we had considered genotypes with only a single read across replicates in our analyses, we could have

mapped the landscape for 78% of the total genotype space (see updated **Supplementary Figure S7**). Some previous studies have indeed included genotypes with single read counts in their analysis^{22–24}, but we adopted a more conservative approach for greater reliability. Given that we have 13 bins and most sequences are typically found within an average of 3.4 bins (see **Supplementary Figure S10**), our stringent read count threshold helps to reduce misestimation of repression levels. We have updated the **Results** section and the **Materials and Methods** (lines 186-192 and lines 763-778, respectively, updated **Supplementary Figure S7**) to better explain the causes of the diversity loss during the filtering process.

Excerpt from Results, lines 186-192:

*We sorted cells harboring our library into 13 “bins” based on their fluorescence, and deep-sequenced TFBS variants from each bin (see **Methods, Supplementary Methods 7-8, Supplementary Figures S5-S6**). The resulting library comprised 48,937 genotypes, covering 75% of the $4^8=65,536$ genotypes within the studied genotype space (**Supplementary Figure S7**). For all further analyses, we additionally applied a stringent sequencing depth threshold. That is, we analyzed only TFBS variants with at least 30 sequencing reads, which reduced our library size further to 17,851 genotypes (27% of 4^8 ; **Methods, Supplementary Figure S7**).*

4) A more in-depth analysis of the resulting network could have been done. Investigating the connectivity of detected nodes could have given insights into whether some regions of the network and, therefore, some closely related combinations of genotypes are missing. This could have been informative on why so many mutations are missing. On the other hand, if the standard deviation of connectivity is low, this means that the space was properly sampled and that the missing mutants come from methodological limitations.

Thank you for expressing this concern, which we addressed by conducting the recommended connectivity analysis and additional analyses to support our findings. This analysis is now discussed in detail in the **Results** section (lines 284-311). Here is the **Results** excerpt (lines 284-311) addressing our analysis:

*Because we had filtered our sequence data for quality, we had obtained repression strength data for only some (approximately 27% of 4^8) genotypes before mapping the landscape. This partial sampling may have introduced systematic biases in landscape topography, particularly in peak assignments^{25,26}. However, several analyses suggest that such sampling biases, if they exist, may not be strong. First, the sampled sequences are not strongly biased by repression strength. Specifically, we analyzed the relative connectivity of genotypes, defined as the fraction of each genotype’s 24 possible neighbors for which we have regulation data (average relative connectivity: 0.34 ± 0.17 per genotype, mean \pm s.d.) This analysis revealed only a weak correlation between connectivity and repression strength ($R = -0.05$, p -value = 2×10^{-14} , **Supplementary Figure S14**), indicating that strongly and weakly repressing genotypes are represented with approximately equal frequency. Furthermore, the relative connectivity of peak genotypes is also only weakly correlated with repression strength ($R = -0.05$, p -value = 2×10^{-14} , **Supplementary Figure S14**). Unrelatedly, we note that the high prevalence of sign epistasis in our landscape (**Supplementary Table S1**) further supports the notion of a highly rugged landscape. In sum, these analyses suggest that our landscape sampling did not have a strong impact on the aspects of landscape topography we study.*

Finally, comparing our observed number of peaks (2,092) with the expected number for a landscape with randomly assigned repression levels reinforces our assessment of high

*ruggedness. To estimate the number of peaks in uncorrelated landscapes, we randomly shuffled the repression strength distribution of our experimentally mapped landscape 1,000 times among the landscape's genotypes. For each of these 1'000 randomized landscapes we then assessed the distribution of the number of peaks (see **Supplementary Methods 9.6**). This resulted in a normal distribution with an average of $2,721 \pm 92$ peaks (mean \pm s.d.), and an upper bound of 2,872 peaks at a 95% confidence level (the upper bound at this confidence level is the value below which 95% of the observations fall, as described in **Supplementary Methods 9.6**). Thus, the number of peaks in the biological landscape is only 24% percent lower than that of randomized landscapes.*

Taken together, these observations support the notion that our sampling did not introduce strong biases with respect to regulation strength in the sampled network.

Additionally, our observation that sign epistasis is prevalent in our landscape (**Table S1**) supports the notion that the ruggedness of our landscape is not a sampling artefact. Assessments of epistasis are based on locally complete information (quadruplets of a genotype, two single mutants and a double mutant) and thus less susceptible to sampling bias (further discussed in **discussion lines 606-616**).

*It [the TetR landscape] is also rugged by another measure, a non-additive interaction between mutations known as reciprocal sign epistasis, in which two individual mutations reduce repression strength but both mutations together increase it (**Supplementary Methods 9.5**)²⁷⁻²⁹. Specifically, 30% of mutant pairs in our landscape exhibit reciprocal sign epistasis (**Supplementary Figure S24, Supplementary Methods 9.5**). This is a much higher incidence than in smooth and better-studied eukaryotic regulatory landscapes, where only 4,5% of mutant interactions in TFBSs exhibit reciprocal sign epistasis¹. Well-studied protein adaptive landscapes also show less reciprocal sign epistasis (8-22%³⁰⁻³³). However, recent studies of epistatic interactions in TFBSs of bacterial and phage TFs (*LacI*^{27,34}, *AraC*¹⁵ and *Cl*¹⁷) have shown that sign epistasis is pervasive in these TFBSs. It occurs in more than half of the studied mutants.^{15,17,27,34}.*

Additionally, in response to a suggestion by reviewer #2, we tested whether our genotype network behaves like a random subgraph of the full Hamming space, which has well-known statistical properties (e.g., work by Stadler and Reidys^{35,36}). We only show the results of this analysis in this reply letter, as it does not materially affect the conclusions of our manuscript. The analysis shows that the observed and expected binomial distributions have very similar means (expected mean = 8.41, observed mean = 8.43) and median (expected median = 8, observed median = 8). However, the degree distributions themselves are substantially different (Figure R1, Chi-squared statistic: 27,651.78, df = 24, p-value < 0.001). Specifically, in the experimental data both lowly and highly connected genotypes are overrepresented compared to what would be expected under a binomial distribution. Importantly, because our other analyses showed that observed node degree and repression strength are only weakly correlated, this deviation from a binomial distribution does not entail a strong sampling bias in the quantities that are important for our analyses.

Figure R1: Comparison of observed and binomial degree distributions in the genotype network. The histogram shows the degree distribution of the observed genotype network (blue, $N=17,765$) compared with the expected binomial distribution (red, $N=17,765$). Despite similar means (observed mean = 8.43, expected mean = 8.41) and medians (observed median = 8, expected median = 8), a Chi-squared test revealed significant differences between the observed and expected distributions (Chi-squared statistic: 27,651.78, $df = 24$, $p\text{-value} < 0.001$). The blue horizontal line represents the mean of the observed degree distribution.

In sum, the weak correlation between connectivity and regulation strength, the prevalence of sign epistasis all support the validity of our landscape characterization. We have incorporated these analyses into the revised manuscript, specifically in new sections added to the **Results** (outlined on lines 284-311), **Discussion** (lines 606-616) and **Supplementary Figure S14**.

5) The notation of +/- s.d. should be standardized. See, for instance, lines 399-400 and 401.

Thank you for pointing out this inconsistency. We have now standardized this notation across the document.

6) The fact that some promoter sequences are stronger repressors than the WT is an interesting observation, but these results are not validated further, and these could stand out just because of noise in the measurements. Are differences statistically significant? Cell sorting is probably noisier in the lower expression bins, so it would be important to validate this result using another approach. It seems that from Figure 2c, the authors are set up to perform validation experiments.

Very good suggestion. In response, we performed the validation experiments you suggested. Specifically, to validate the regulatory strength differences observed in the lowest bins (for the strongest repression), we measured the expression levels of 45 sequences from these bins using a plate reader and compared them to the WT. The selected sequences exhibit substantially

higher repression strength than the WT, with a relative repression increase of $14.6\% \pm 3.2\%$ (mean \pm s.d.). This difference is statistically significant (Welch one-sample t-test, $t = -55.737$, $df = 44$, $p\text{-value} < 2.2 \times 10^{-16}$). The results from these new experiments support our original observations and are now discussed on **lines 205-219** and presented in **Supplementary Figure S12**).

7) The measure of reproducibility is presented as a correlation of read counts of each variant (Figure S7). While this is useful, it just represents technical reproducibility in the library construction process. What is critical is the correlation between the three replicates in the estimated repression levels of individual variants, i.e. in the measures that are used to make inferences on the adaptive network. It would also be important to know how measurement errors impact the overall structure of the network, the identity of the peaks and, for instance see point 6 above. Experimental noise is likely not equally distributed among the expression bins, so is it possible that measurement noise itself impacts the navigability estimates of the landscape?

Your observation is entirely correct. The measure of reproducibility presented in **Figure S7** focuses on the correlation of read counts for each variant, which indeed reflects technical reproducibility during the library construction process. While it is a useful metric, we agree that it does not fully capture the reproducibility in the repression levels of individual variants.

To address this concern, we have analyzed the Pearson correlation between the three replicates in the estimated repression levels, which are important for making inferences about the regulatory landscape. We now added this comparison as **Supplementary Figure S11** and mention it in the results (**lines 205-209**), as follows.

Fluorescence-based sorting methods are inherently noisy, especially at the highest and lowest bounds of the fluorescence distribution³⁷⁻³⁹. To assess how such variability could impact our binding strength estimates, we measured the correlation of our repression strength measurements between replicates. We found that the Pearson's correlation coefficients for replicates were high, ranging from 0.84 to 0.92 (Supplementary Figure S11).

We also agree with the reviewer that measurement errors could impact our landscape's structure and navigability^{1,40}. We thus performed a new analysis to address this point and now discuss its results in the paper (**lines 510-528**, and a new section in **Supplementary Methods 11**). In part, this discussion reads as follows:

*Lastly, because gene expression and sort-seq experiments are noisy³⁷⁻³⁹, we asked how such noise could influence landscape topography and adaptive evolution on the landscape^{1,33,40}. To this end, we estimated an individual experimental noise value (τ) for the repression strength S_A and S_B for each focal genotype A , and for each of its single mutants B , based on the standard deviation of their repression strengths across our three replicate experiments (**Supplementary Methods 11**). We then considered the repression strength S_B of a mutant to be indistinguishable from S_A if its average across the replicates lies within the experimental noise of S_A (**Supplementary Methods 11**). We mapped the regulatory landscape under this assumption for varying values of this noise (**Supplementary Methods 11**). Not surprisingly, as noise increases, the landscape becomes smoother; the number of peaks decreases, and peaks merge into increasingly broader plateaus (**Supplementary Figure S21**). Accessibility of these peaks also increases (**Supplementary Figure S21**), which is reflected by an increasing proportion (25%*

to 37%) of adaptive walks that reach at least one high repression peak (**Supplementary Figure S22**).

*In addition, we observed that the number of times a genotype is visited during an adaptive walk is not highly sensitive to noise (**Supplementary Figure S23**). This is relevant because it shows that regardless of landscape changes caused by noise, adaptive walks traverse the same genotypes with similar frequency, indicating similar evolutionary dynamics.*

8) It was difficult to understand the role of the PCA analysis in the paper and what it represents, especially given that the top two PCs explain only 5% of the variation. Also, the sequence logo in Figure 2d seems to confirm this, with one preferred nucleotide at each position and most alternatives generally not tolerated.

The point is well taken. Relatedly, reviewer #2 correctly pointed out that it is difficult to judge the clustering of the peaks based on only two principal components (PCs), which explain only a small proportion of the genetic variation. The distribution of genetic distances between peaks (**Figs. 3b and 3c**) provides a more direct demonstration that the peaks are not highly clustered.

Part of the interpretation problems with the PCA analysis is that the sequence space underlying our landscape is high-dimensional. To clarify this, we now included an explanation in **Supplementary Methods 9.7 (lines 588-599)** with projections on the first five components of the PCA (**Supplementary Figure S16**).

*The sequence space underlying our landscape is high-dimensional. Like other spaces in which DNA sequences can harbor more than two alleles (A,T,G,C) per site, it forms a Hamming graph⁴¹. In contrast, spaces with two alleles per site form a simpler topology called a hypercube, a special case of a Hamming graph⁴¹. Although dimensionality reduction techniques like PCA are applied in the visualization of genotype spaces⁴²⁻⁴⁴, their effectiveness in representing combinatorially complex spaces, such as Hamming graphs, can be limited due to minimal redundancy in genetic information across different dimensions⁴⁵. In consequence, each principal component explains only a small amount of genetic variation. For example, PC1 explains only 5% of the variance in our data. Another consequence is that the clustering of peaks may depend on whether they share some allelic states. Thus, we now include projections on the first five components of the PCA to illustrate such different clustering (**Supplementary Figure S16**).*

9) I could not see in the simulation data the different types of mutations (transition and transversions). As far as I understand, for now, mutational neighbours are just one nucleotide change away. However, if mutations do not have the same rates, this could change conclusions regarding the navigability of this network.

We are clearly at fault here for not specifying how we modelled mutations. We performed all evolutionary simulations with mutational biases specific to *E. coli*, as quantified by Lind and Andersson (2008)⁴⁶. We now mention this in **Supplementary Methods 10 (lines 602-607)**.

We simulated the adaptive evolution of a population on the adaptive landscape by performing three different types of random walks, a “greedy adaptive” random walk, a “uniform adaptive” random walk, and a random walk based on Kimura’s model of fixation probability. Our simulations also account for mutation bias, which means that different types of mutation have

*a different probability of occurring in a population*⁴⁷. *We use mutation biases that were experimentally determined for Escherichia coli*^{46,48}.

Minor comments:

Line 62: “The molecular origins and evolution of such elements are still poorly understood.” This sounds like a very strong statement.

We agree with the reviewer and have corrected the statement as follows (**lines 61-69**) for a broader contextualization:

Despite numerous theoretical studies exploring the molecular origins and evolution of these elements^{49–58}, *comprehensive and high-throughput empirical research addressing these questions only began about a decade ago*^{3,22–24,59–67}. *Additionally, while extensive in vitro data on the binding of eukaryotic TFs to their binding sites have advanced our understanding of eukaryotic genotype-phenotype maps of gene regulation*^{1,3,68–72}, *the topography of these landscapes remains largely unexplored in prokaryotes. The few studies that do examine the relationships between TFBS sequences and gene regulation in bacteria primarily focus on the mechanisms of gene regulation rather than the evolution of gene regulation*^{24,60,62,73–76}.

Reviewer #2 (Remarks to the Author):

The authors present a large genotype-phenotype map that relates an 8-letter subsequence of a bacterial transcription factor binding site to the repression level induced by the tetracycline repressor TetR. The resulting landscape is analyzed with respect to its ruggedness (number and distribution of peaks) and navigability (existence of monotonic paths to peaks, basins of attraction of peaks, reachability of peaks by various types of evolutionary dynamics). The methodology appears to be sound, and the methods and results are clearly described. There are few (if any) data sets of comparable size and quality in the current literature, and the work therefore constitutes an important advance in the field that is principally well suited for publication in Nature Communications. However, I have some concerns about the analysis and interpretation of the results that the authors should address before I can recommend publication.

MAJOR CONCERNS

1. The studied landscape comprises only about 27% of the 65536 possible sequences. Since the TetR repression level does not affect the viability of the bacteria, my understanding is that this should be considered as a random, unbiased sample of the full landscape, the sparsity of which is mainly determined by the chosen sequencing depth (Fig. S8). However, the authors do not present any evidence that the sample is indeed unbiased. If it is, the resulting genotype network should be a random subgraph of the full Hamming space. The statistical properties of such subgraphs are well known in computational biology (see, e.g. work by Stadler and Reidys). For example, the degree distribution is expected to be binomial. The authors should analyze the structure of their genotype network along these lines, and if possible also study the effects of varying the sequencing depth threshold (and thus the sparsity of the network).

Thank you for this comment and for the excellent suggestion. Although we cannot prove with absolute certainty that our sampling is unbiased, we have followed your suggestion and a related one by reviewer 1 by performing several new analyses. Please see **Figure R1** in our response to **item 4** of **reviewer 1** and the accompanying text, which shows the results of your suggested analysis on the degree distribution of the sampled graph, as well as that of other analyses.

To discuss these analyses, we added the following whole paragraph to the results section (**lines 284-299**).

*Because we had filtered our sequence data for quality, we had obtained repression strength data for only some (approximately 27% of 4^8) genotypes before mapping the landscape. This partial sampling may have introduced systematic biases in landscape topography, particularly in peak assignments^{25,26}. However, several analyses suggest that such sampling biases, if they exist, may not be strong. First, the sampled sequences are not strongly biased by repression strength. Specifically, we analyzed the relative connectivity of genotypes, defined as the fraction of each genotype's 24 possible neighbors for which we have regulation data (average relative connectivity: 0.34 ± 0.17 per genotype, mean \pm s.d.) This analysis revealed only a weak correlation between connectivity and repression strength ($R = -0.05$, p -value = 2×10^{-14} , **Supplementary Figure S14**), indicating that strongly and weakly repressing genotypes are represented with approximately equal frequency. Furthermore, the relative connectivity of peak genotypes is also only weakly correlated with repression strength ($R = -0.05$, p -value = 2×10^{-14} , **Supplementary Figure S14**). Unrelatedly, we note that the high prevalence of sign epistasis in our landscape (**Supplementary Table S1**) further supports the notion of a highly rugged*

landscape. In sum, these analyses suggest that our landscape sampling did not have a strong impact on the aspects of landscape topography we study.

2. The fact that the landscape is a (sparse) random sample has important implications for the interpretation of the results, in particular for the identification and analysis of peaks. There are two opposing effects. On the one hand, a large fraction (73%) of peaks are not sampled. On the other hand, genotypes are falsely identified as peaks, because they have neighbors of higher repression level that are not sampled. It is possible (and can be shown for uncorrelated random landscapes) that these effect partially cancel, such that the number of (mis-)identified peaks is similar to the number of peaks of the full landscape; indeed, the expected number of peaks for the full landscape with randomly assigned repression levels is 2621 according to a formula first derived by Kauffman and Levin (JTB 1987), not much larger than the number reported here (2092).

To put their estimate for the number of peaks into perspective, the authors could simply place uncorrelated random numbers onto the measured genotype network and determine the number of peaks of this random landscape. This should provide an upper bound on the ruggedness (compare to Ref.36). I expect that this bound will not be much larger than their actual estimate, supporting and quantifying their claim that the measured landscape is highly rugged. It would also be of interest to classify the genotypes identified as peaks by the number of neighboring genotypes that belong to the network, in order to determine how many are true peaks of the full landscape, and whether these tend to have high expression levels.

Thank you for your suggestion. Indeed, we found a number of peaks that is not far below that expected in a maximally rugged landscape. We have added this observation to the new paragraph in the results discussing the sparsity of our network (**lines 301-311**).

*Finally, comparing our observed number of peaks (2,092) with the expected number for a landscape with randomly assigned repression levels reinforces our assessment of high ruggedness. To estimate the number of peaks in uncorrelated landscapes, we randomly shuffled the repression strength distribution of our experimentally mapped landscape 1,000 times among the landscape's genotypes. For each of these 1'000 randomized landscapes we then assessed the distribution of the number of peaks (see **Supplementary Methods 9.6**). This resulted in a normal distribution with an average of $2,721 \pm 92$ peaks (mean \pm s.d.), and an upper bound of 2,872 peaks at a 95% confidence level (the upper bound at this confidence level is the value below which 95% of the observations fall, as described in **Supplementary Methods 9.6**). Thus, the number of peaks in the biological landscape is only 24% percent lower than that of randomized landscapes.*

3. The authors argue that the fact that high repression peaks are more accessible than low peaks is "striking" and exemplifies a property "typically associated with smooth landscapes" (lines 524-527). I disagree with this statement, and actually believe it would be hard to come up with a fitness landscape where peak accessibility (measured in terms of quantities like the size of the basin of attraction or the number of accessible paths) is not correlated with peak height. In particular, such a correlation is well documented and understood for the null model with uncorrelated random fitness landscapes [often referred to as the House-of-Cards (HoC) model], which would not typically be considered a model with high navigability. Recent work on HoC landscapes has shown that the probability of existence of an accessible path shifts sharply from

0 to 1 at a (computable) threshold value of the fitness difference between the initial and final genotype [see Schmiegelt and Krug, J. Math. Biol. 2023, as well as arXiv:1903.11913]. The threshold depends on the distance between the genotypes (scaled with the length of the genetic sequence) but remains nontrivial even at maximal distance. This implies that the fraction of genotypes from which a given peak genotype can be accessed increases continuously with the height of the peak, once the latter is large enough. The correlation between peak height and accessibility is therefore highly generic, and as such does not necessitate a "refinement of current landscape theory" (line 580/581).

Thank you for your insightful feedback regarding the correlation between peak height and accessibility in fitness landscapes. We especially appreciate your reference to the House-of-Cards (HoC)⁷⁷ model and the recent work by Schmiegelt and Krug⁷⁸, which indeed illustrates that in uncorrelated random fitness landscapes, peak accessibility is correlated with peak height. In response, we have rewritten an entire passage of the discussion to acknowledge this literature (**lines 685-696**). It reads as follows:

Our work also shows that high repression peaks are more accessible, which aligns with theoretical predictions from simple theoretical models of adaptive landscapes. Specifically, in the House-of-Cards (HoC) model where neighboring genotypes have uncorrelated fitness values^{77,79}, peak accessibility correlates with peak height^{25,78}. More generally, Das et al. (2020)⁸⁰ described a novel category of rugged yet highly navigable landscapes, which shows that empirical fitness landscapes can display properties not typically predicted by simple statistical models⁸¹⁻⁸³. Schmiegelt and Krug⁷⁸ also noted that the HoC model's assumption of independent and identically distributed random fitness values does not reflect the varying degrees of fitness correlations seen in empirical landscapes. In this context, our study underscores the complexity of biological data. It also highlights the importance of empirical studies in validating theoretical models, and motivating future theoretical work that hews close to experimental data.

OTHER ISSUES

4. line 166: "absence of the inducer". It is not mentioned in the main text that the "inducer" is Atc. This should be explained here or in the caption of Figure 1.

Corrected. We now mention the inducer in **Figure 1b** and on **line 179**.

5. Figure 3d, lines 265-268: One of the two clouds appears to contain more peaks than the other. Which one is it?

We now explicitly describe which cloud has more peaks in **the results section (lines 332-333)** and in the caption of **figure 3d**.

Excerpt from **lines 332-333**

We observed that 75% of peaks and 63% of high peaks fall into this second cloud (Figure 3d).

6. Figure 4c: The grey shade around the black curve is hard to see.

Corrected. We changed the shade color to green to improve visibility.

7. Figure 4f: I don't quite understand this panel. How are the genotypes ordered in this plot? What is the origin of the blue cross? Also, in the caption it is stated that a value of zero is represented in yellow color, whereas in fact it appears to be blue.

Corrected. Rows and columns were clustered by a hierarchical clustering method with complete linkage using the *heatmap* R package⁸⁴. The blue cross represents the clustering of pairs of basins of attraction with no overlap. We have added this information to the legend of **figure 4f** and corrected the color schemes in the caption.

8. line 380, Figure 5: What is meant by "all possible paths (both accessible and inaccessible)"? The total number of all paths between two genotypes (direct and indirect) grows double exponentially with genetic distance, and even the number of direct paths grows faster than exponentially (for two genotypes at distance d , it is $d!$). For $d=8$ there are 40320 direct paths, which is off the scale of Fig. 5b. Please clarify.

Thank you for pointing this out. In our study we are working with a directed genotype network, where a directed edge points from a genotype (A) to a one-mutant neighbor (B), if the repression strength of genotype B is considered to be higher than the one of genotype A (see **Supplementary Methods 11**).

As discussed before, this network represents a sampled subset of a larger complete genotype network (4^8) due to the absence of many genotypes from our quality-filtered data.

Accessible paths in this context are those that exist within the directed network, leading from a starting genotype to a high peak through single mutations encompassing increasing regulatory strengths. When we refer to "all possible paths," we are ignoring the network's directionality, meaning that we consider all paths to peaks regardless of directionality. We now clarify this distinction in the **results section** (lines **449-452**).

*To find out whether this is the case, we enumerated all paths, both accessible and inaccessible, from every non-peak genotype to every peak genotype (**Supplementary Methods 9.5**). Our analysis revealed an exponential increase in the number of paths to a peak as the genetic distance between the variant and the peak increased (**Figure 5b**).*

It is important to note that the factorial growth ($d!$) of direct paths only applies to a combinatorially complete network. Our network, however, is incomplete due to the unsampled genotypes. Consequently, even if there is a genetic distance of 8 base pairs between a genotype and a peak, the actual number of available paths is substantially reduced. We appreciate your attention to this detail, and we have ensured that the **Figure 5b** legend (lines **554-564**) more clearly reflect this nuance:

b. The landscape harbors many paths leading to high repression peaks. *The vertical axis shows the total number of shortest paths per variant to a high peak, while the horizontal axis depicts the length of these shortest paths. Red and blue boxplots summarize the number of all shortest paths and of all accessible shortest paths, respectively. Each box represents the interquartile range (IQR), with the horizontal line inside each box indicating the median value. Whiskers span the interval defined by $1.5 \times \text{IQR}$, with data values beyond this range not shown. The total number of paths is $N = 47,950,663$. We note that for a combinatorially complete landscape, the number of direct paths would be expected to grow factorially with genetic*

distance ($d!$), i.e., there are 40,320 paths for genotypes separated by a genetic distance 8 mutations. However, due to the incompleteness of our landscape, the number of paths we observe is lower.

9. lines 421-428: How are the selection coefficients computed when evaluating the Kimura fixation probability? Is the normalized repression level simply interpreted as fitness? If so, this should be said.

Corrected. We expanded the description of our simulations, explicitly describing how we computed the selection coefficients in the supplementary material (**Supplementary Methods 10**), and highlighting that we use the normalized repression level as a proxy for fitness in both the main text (**lines 492-495**) and in the revised **Supplementary Methods 10 (lines 625-633)**. In the main text, **lines 491-495** read as follows:

*We simulated adaptive walks with an approach pioneered by Kimura^{85,86} that allows us to calculate the fixation probability of any mutation as a function of population size and the selection coefficient s , i.e., the difference in repression strength between neighboring genotypes^{85,87}. (**Supplementary Methods 10**).*

10. line 465: assessed -> accessed? In the same line, the reference to Figure 5b appears to be incorrect.

Corrected

11. Figure 5d, Figs. S15, S16: All these figures show a discontinuity in the cumulative frequency of adaptive walks around the wild type repression level. Where does this come from?

The discontinuity in the cumulative frequency distribution comes from a discontinuity in the distribution of repression strength values around that region; this can also be observed in the distribution of repression strengths (**Figure 2b**). We now clarify this observation in the figure legend.

12. Some references to the bibliography appear to be incorrect. For example, on line 237, Ref.51 (Blount et al.) does not seem to be the correct reference for the giant component of a network, and on line 607 of the Supplement a reference 108 is cited that does not exist.

Thank you for catching this. We have corrected the references.

REFERENCES:

1. Aguilar-Rodríguez, J., Payne, J. L. & Wagner, A. A thousand empirical adaptive landscapes and their navigability. *Nat Ecol Evol* **1**, 0045 (2017).
2. Mustonen, V., Kinney, J., Callan, C. G. & Lassig, M. Energy-dependent fitness: A quantitative model for the evolution of yeast transcription factor binding sites. *Proceedings of the National Academy of Sciences* (2008) doi:10.1073/pnas.0805909105.
3. Haldane, A., Manhart, M. & Morozov, A. V. Biophysical Fitness Landscapes for Transcription Factor Binding Sites. *PLoS Comput Biol* **10**, 36–38 (2014).
4. Mustonen, V. & Lassig, M. Evolutionary population genetics of promoters: Predicting binding sites and functional phylogenies. *Proceedings of the National Academy of Sciences* **102**, 15936–15941 (2005).
5. Weinert, F. M., Brewster, R. C., Rydenfelt, M., Phillips, R. & Kegel, W. K. Scaling of gene expression with transcription-factor fugacity. *Phys Rev Lett* **113**, 1–5 (2014).
6. Garcia, H. G. *et al.* Operator sequence alters gene expression independently of transcription factor occupancy in bacteria. *Cell Rep* **2**, 150–161 (2012).
7. Razo-Mejia, M. *et al.* Tuning Transcriptional Regulation through Signaling: A Predictive Theory of Allosteric Induction. *Cell Syst* **6**, 456–469.e10 (2018).
8. Srivastava, M. & Payne, J. L. On the incongruence of genotype-phenotype and fitness landscapes. *PLoS Comput Biol* **18**, e1010524 (2022).
9. Shultzaberger, R. K., Malashock, D. S., Kirsch, J. F. & Eisen, M. B. The Fitness Landscapes of cis-Acting Binding Sites in Different Promoter and Environmental Contexts. *PLoS Genet* **6**, e1001042 (2010).
10. Ellinger, T., Behnke, D., Bujard, H. & Gralla, J. D. Stalling of Escherichia coli RNA polymerase in the +6 to +12 region in vivo is associated with tight binding to consensus promoter elements. *J Mol Biol* **239**, 455–465 (1994).
11. Berens, C. & Hillen, W. Gene regulation by tetracyclines. Constraints of resistance regulation in bacteria shape TetR for application in eukaryotes. *Eur J Biochem* **270**, 3109–3121 (2003).
12. Melnyk, A. H., Wong, A. & Kassen, R. The fitness costs of antibiotic resistance mutations. *Evol Appl* (2015) doi:10.1111/eva.12196.
13. LENSKI, R. E. *et al.* Epistatic effects of promoter and repressor functions of the Tn10 tetracycline-resistance operon on the fitness of Escherichia coli. *Mol Ecol* **3**, 127–135 (1994).
14. Nguyen, T. N. M., Phan, Q. G., Duong, L. P., Bertrand, K. P. & Lenski, R. E. Effects of carriage and expression of the Tn10 tetracycline-resistance operon on the fitness of Escherichia coli K12. *Mol Biol Evol* **6**, 213–225 (1989).
15. Lagator, M., Iglér, C., Moreno, A. B., Guet, C. C. & Bollback, J. P. Epistatic interactions in the arabinose cis-regulatory element. *Mol Biol Evol* **33**, 761–769 (2016).
16. de Vos, M. G. J., Poelwijk, F. J., Battich, N., Ndika, J. D. T. & Tans, S. J. Environmental Dependence of Genetic Constraint. *PLoS Genet* **9**, e1003580 (2013).
17. Lagator, M., Paixão, T., Barton, N. H., Bollback, J. P. & Guet, C. C. On the mechanistic nature of epistasis in a canonical cis-regulatory element. *Elife* **6**, 1–16 (2017).

18. de Visser, J. A. G. M., Cooper, T. F. & Elena, S. F. The causes of epistasis. *Proceedings of the Royal Society B: Biological Sciences* **278**, 3617–3624 (2011).
19. Crocker, J., Preger-Ben Noon, E. & Stern, D. L. *The Soft Touch: Low-Affinity Transcription Factor Binding Sites in Development and Evolution. Current Topics in Developmental Biology* vol. 117 (Elsevier Inc., 2016).
20. Eckert, B. & Beck, C. F. Overproduction of transposon Tn10-encoded tetracycline resistance protein results in cell death and loss of membrane potential. *J Bacteriol* **171**, 3557–3559 (1989).
21. Rajer, F. & Sandegren, L. The Role of Antibiotic Resistance Genes in the Fitness Cost of Multiresistance Plasmids. *mBio* **13**, (2022).
22. de Boer, C. G. *et al.* Deciphering eukaryotic gene-regulatory logic with 100 million random promoters. *Nature Biotechnology* 2019 38:1 **38**, 56–65 (2019).
23. Vaishnav, E. D. *et al.* The evolution, evolvability and engineering of gene regulatory DNA. *Nature* 2022 603:7901 **603**, 455–463 (2022).
24. Kinney, J. B., Murugan, A., Callan, C. G. & Cox, E. C. Using deep sequencing to characterize the biophysical mechanism of a transcriptional regulatory sequence. *Proceedings of the National Academy of Sciences* **107**, 9158–9163 (2010).
25. Krug, J. & Oros, D. Evolutionary accessibility of random and structured fitness landscapes. (2023).
26. De Visser, J. A. G. M. & Krug, J. Empirical fitness landscapes and the predictability of evolution. *Nat Rev Genet* **15**, 480–490 (2014).
27. Poelwijk, F. J., Tănase-Nicola, S., Kiviet, D. J. & Tans, S. J. Reciprocal sign epistasis is a necessary condition for multi-peaked fitness landscapes. *J Theor Biol* **272**, 141–144 (2011).
28. Saona, R., Kondrashov, F. A. & Khudiakova, K. A. Relation Between the Number of Peaks and the Number of Reciprocal Sign Epistatic Interactions. *Bull Math Biol* **84**, (2022).
29. Kvittek, D. J. & Sherlock, G. Reciprocal sign epistasis between frequently experimentally evolved adaptive mutations causes a rugged fitness landscape. *PLoS Genet* (2011) doi:10.1371/journal.pgen.1002056.
30. Li, C., Qian, W., Maclean, C. J. & Zhang, J. The fitness landscape of a tRNA gene. *Science* (1979) (2016) doi:10.1126/science.aae0568.
31. Li, C. & Zhang, J. Multi-environment fitness landscapes of a tRNA gene. *Nature Ecology & Evolution* 2018 2:6 **2**, 1025–1032 (2018).
32. Papkou, A., Garcia-Pastor, L., Escudero, J. A. & Wagner, A. A rugged yet easily navigable fitness landscape of antibiotic resistance. *bioRxiv* 2023.02.27.530293 (2023) doi:10.1101/2023.02.27.530293.
33. Song, S. & Zhang, J. Unbiased inference of the fitness landscape ruggedness from imprecise fitness estimates. *Evolution (N Y)* **75**, 2658–2671 (2021).
34. Poelwijk, F. J., Kiviet, D. J., Weinreich, D. M. & Tans, S. J. Empirical fitness landscapes reveal accessible evolutionary paths. *Nature* (2007) doi:10.1038/nature05451.
35. Reidys, C. M. & Stadler, P. F. Combinatorial Landscapes. *SIAM Review* **44**, 3–54 (2002).
36. Reidys, C. M. & Stadler, P. F. Neutrality in fitness landscapes. *Appl Math Comput* **117**, 321–350 (2001).

37. Peterman, N. & Levine, E. Sort-seq under the hood: Implications of design choices on large-scale characterization of sequence-function relations. *BMC Genomics* **17**, 1–17 (2016).
38. Trippe, B. L. *et al.* Randomized gates eliminate bias in sort-seq assays. *Protein Science* **31**, e4401 (2022).
39. Gilliot, P. A. & Gorochowski, T. E. Effective design and inference for cell sorting and sequencing based massively parallel reporter assays. *Bioinformatics* **39**, (2023).
40. Levitan, B. & Kauffman, S. Adaptive walks with noisy fitness measurements. *Mol Divers* **1**, 53–68 (1995).
41. Stadler, P. F. & Stadler, B. M. R. Genotype-Phenotype Maps. *Biol Theory* **1**, 268–279 (2006).
42. Jombart, T., Devillard, S., Dufour, A. B. & Pontier, D. Revealing cryptic spatial patterns in genetic variability by a new multivariate method. *Heredity (Edinb)* **101**, (2008).
43. Privé, F., Luu, K., Blum, M. G. B., McGrath, J. J. & Vilhjálmsson, B. J. Efficient toolkit implementing best practices for principal component analysis of population genetic data. *Bioinformatics* **36**, (2020).
44. Papkou, A., Garcia-Pastor, L., Escudero, J. A. & Wagner, A. A rugged yet easily navigable fitness landscape. *Science (1979)* **382**, (2023).
45. Rowe, W. *et al.* Analysis of a complete DNA-protein affinity landscape. *J R Soc Interface* **7**, 397–408 (2010).
46. Lind, P. A. & Andersson, D. I. Whole-genome mutational biases in bacteria. *Proceedings of the National Academy of Sciences* **105**, 17878–17883 (2008).
47. Cano, A. V. *et al.* Mutation bias and the predictability of evolution. *Philosophical Transactions of the Royal Society B* **378**, (2023).
48. Horton, J. S. & Taylor, T. B. Mutation bias and adaptation in bacteria. *Microbiology (N Y)* **169**, (2023).
49. Berg, J., Willmann, S. & Lässig, M. Adaptive evolution of transcription factor binding sites. *BMC Evol Biol* **4**, 564–567 (2004).
50. Kotelnikova, E. A., Makeev, V. J. & Gelfand, M. S. Evolution of transcription factor DNA binding sites. *Gene* **347**, 255–263 (2005).
51. Moses, A. M., Chiang, D. Y., Kellis, M., Lander, E. S. & Eisen, M. B. Position specific variation in the rate of evolution in transcription factor binding sites. *BMC Evol Biol* **3**, 1–13 (2003).
52. Hahn, M. W. The Effects of Selection Against Spurious Transcription Factor Binding Sites. *Mol Biol Evol* **20**, 901–906 (2003).
53. Mrázek, J. & Karls, A. C. In silico simulations of occurrence of transcription factor binding sites in bacterial genomes. *BMC Evol Biol* **19**, 1–12 (2019).
54. Tuğrul, M., Paixão, T., Barton, N. H. & Tkačik, G. Dynamics of Transcription Factor Binding Site Evolution. *PLoS Genet* **11**, 1–28 (2015).
55. Gorbunov, K. Y., Laikova, O. N., Rodionov, D. A., Gelfand, M. S. & Lyubetsky, V. A. Evolution of regulatory motifs of bacterial transcription factors. *In Silico Biol* **10**, 163–183 (2010).
56. Kurafeiski, J. D., Pinto, P. & Bornberg-Bauer, E. Evolutionary potential of cis-regulatory mutations to cause rapid changes in transcription factor binding. *Genome Biol Evol* **11**, 406–414 (2019).

57. Babu, M. M. Structure, evolution and dynamics of transcriptional regulatory networks. *Biochem Soc Trans* **38**, 1155–1178 (2010).
58. Majic, P. & Payne, J. L. Enhancers Facilitate the Birth of De Novo Genes and Gene Integration into Regulatory Networks. *Mol Biol Evol* **37**, 1165–1178 (2020).
59. Rockel, S., Geertz, M. & Maerkl, S. J. MITOMI: A microfluidic platform for in vitro characterization of transcription factor–DNA interaction. *Methods in Molecular Biology* (2012) doi:10.1007/978-1-61779-292-2_6.
60. Belliveau, N. M. *et al.* A Systematic and Scalable Approach for Dissecting the Molecular Mechanisms of Transcriptional Regulation in Bacteria. *Biophys J* **114**, 151a (2018).
61. Kinney, J. B. & McCandlish, D. M. Massively Parallel Assays and Quantitative Sequence–Function Relationships. *Annu Rev Genomics Hum Genet* **20**, annurev-genom-083118-014845 (2019).
62. Barnes, S. L., Belliveau, N. M., Ireland, W. T., Kinney, J. B. & Phillips, R. Mapping DNA sequence to transcription factor binding energy in vivo. *PLoS Comput Biol* **15**, e1006226 (2019).
63. Vaknin, I. & Amit, R. Molecular and experimental tools to design synthetic enhancers. *Curr Opin Biotechnol* **76**, 102728 (2022).
64. Fuqua, T. *et al.* Dense and pleiotropic regulatory information in a developmental enhancer. *Nature* **587**, 235–239 (2020).
65. Kvon, E. Z. *et al.* Comprehensive In Vivo Interrogation Reveals Phenotypic Impact of Human Enhancer Variants. *Cell* **180**, 1262–1271.e15 (2020).
66. Uebbing, S. *et al.* Massively parallel discovery of human-specific substitutions that alter enhancer activity. *Proc Natl Acad Sci U S A* **118**, e2007049118 (2021).
67. Melnikov, A. *et al.* Systematic dissection and optimization of inducible enhancers in human cells using a massively parallel reporter assay. *Nat Biotechnol* **30**, 271–277 (2012).
68. Payne, J. L. & Wagner, A. The Robustness and Evolvability of Transcription Factor Binding Sites. *Science (1979)* **343**, (2014).
69. Schweizer, G. & Wagner, A. Both Binding Strength and Evolutionary Accessibility Affect the Population Frequency of Transcription Factor Binding Sequences in *Arabidopsis thaliana*. *Genome Biol Evol* **13**, (2021).
70. Aguilar-Rodríguez, J. & Payne, J. L. Robustness and Evolvability in Transcriptional Regulation. *Evolutionary Systems Biology* 197–219 (2021) doi:10.1007/978-3-030-71737-7_9.
71. Le, D. D. *et al.* Comprehensive, high-resolution binding energy landscapes reveal context dependencies of transcription factor binding. *Proceedings of the National Academy of Sciences* **115**, 201715888 (2018).
72. Maerkl, S. J. S. J. & Quake, S. R. A systems approach to measuring the binding energy landscapes of transcription factors. *Science (1979)* **315**, (2007).
73. Antunes, L. C. M., Ferreira, R. B. R., Lostroh, C. P. & Greenberg, E. P. A mutational analysis defines *Vibrio fischeri* LuxR binding sites. *J Bacteriol* **190**, 4392–4397 (2007).
74. Ireland, W. T. *et al.* Deciphering the regulatory genome of *Escherichia coli*, one hundred promoters at a time. *Elife* **9**, 1–76 (2020).
75. Lagator, M. *et al.* Predicting bacterial promoter function and evolution from random sequences. *Elife* **11**, (2022).

76. Urtecho, G., Tripp, A. D., Insigne, K. D., Kim, H. & Kosuri, S. Systematic Dissection of Sequence Elements Controlling σ 70 Promoters Using a Genomically Encoded Multiplexed Reporter Assay in *Escherichia coli*. *Biochemistry* **58**, 1539–1551 (2019).
77. Kingman, J. F. C. A simple model for the balance between selection and mutation. *J Appl Probab* **15**, 1–12 (1978).
78. Schmiegelt, B. & Krug, J. Accessibility percolation on Cartesian power graphs. *J Math Biol* **86**, 1–43 (2023).
79. Kauffman, S. & Levin, S. Towards a general theory of adaptive walks on rugged landscapes. *J Theor Biol* **128**, 11–45 (1987).
80. Das, S. G., Direito, S. O. L., Waclaw, B., Allen, R. J. & Krug, J. Predictable properties of fitness landscapes induced by adaptational tradeoffs. *Elife* **9**, (2020).
81. Neidhart, J., Szendro, I. G. & Krug, J. Adaptation in Tunably Rugged Fitness Landscapes: The Rough Mount Fuji Model. *Genetics* **198**, 699–721 (2014).
82. Szendro, I. G., Schenk, M. F., Franke, J., Krug, J. & De Visser, J. A. G. M. Quantitative analyses of empirical fitness landscapes. *Journal of Statistical Mechanics: Theory and Experiment* **2013**, P01005 (2013).
83. Hwang, S., Schmiegelt, B., Ferretti, L. & Krug, J. Universality Classes of Interaction Structures for NK Fitness Landscapes. *J Stat Phys* **172**, 226–278 (2018).
84. Raivo Kolde. pheatmap: Pretty Heatmaps. R package version 1.0.12. <https://CRAN.R-project.org/package=pheatmap> Preprint at (2019).
85. KIMURA, M. ON THE PROBABILITY OF FIXATION OF MUTANT GENES IN A POPULATION. *Genetics* **47**, 713–719 (1962).
86. Crow, J. and Kimura, M. *An Introduction to Population Genetics Theory* [Paperback]. 608 (2009).
87. Kimura, M. *The Neutral Theory of Molecular Evolution*. (Cambridge University Press, 1983). doi:10.1017/CBO9780511623486.

REVIEWERS' COMMENTS

Reviewer #1 (Remarks to the Author):

The authors have addressed all of my comments. Congrats on this nice piece of work!

A: Thank you very much. We truly appreciate your feedback, which substantially contributed to improving our manuscript.

Reviewer #2 (Remarks to the Author):

The issues raised in my previous report have been fully addressed, and I recommend publication of the manuscript in its present form. There are three very minor points that should be fixed prior to publication:

1. In the caption of Figure 5b, I'm a bit puzzled by the information about the total number of paths ("The total number of paths is $N = 47,950,663$ "). Do I understand correctly that this is the number of paths of any length, originating at some variant and reaching some peak? If so, this should be specified more clearly. But in fact I don't really see the importance of this information; it may as well be omitted.

A: We agree with the reviewer that the total number of paths does not add critical information to the caption. We have removed this detail in the revised version for clarity.

2. There is a typo in the caption of Figure 3 (nom-peak -> non-peak).

A: Thank you for catching that error. We have corrected the typo in the caption.

3. Ref. 85 has meanwhile been published (J. Stat. Mech. (2024) 034003).

A: Thank you for informing us. We have updated the reference accordingly.